# Learning to Understand:
# Identifying Interactions via the Möbius Transform

**Justin Singh Kang**
UC Berkeley
justin_kang@berkeley.edu

**Yigit Efe Erginbas**
UC Berkeley
erginbas@berkeley.edu

**Landon Butler**
UC Berkeley
landonb@berkeley.edu

**Ramtin Pedarsani**
UC Santa Barbara
ramtin@ece.ucsb.edu

**Kannan Ramchandran**
UC Berkeley
kannanr@berkeley.edu

## Abstract

One of the key challenges in machine learning is to find interpretable representations of learned functions. The Möbius transform is essential for this purpose, as its coefficients correspond to unique *importance scores* for *sets of input variables*. This transform is closely related to widely used game-theoretic notions of importance like the *Shapley* and *Bhanzaf value*, but it also captures crucial higher-order interactions. Although computing the Möbius Transform of a function with $n$ inputs involves $2^n$ coefficients, it becomes tractable when the function is *sparse* and of *low degree* as we show is the case for many real-world functions. Under these conditions, the complexity of the transform computation is significantly reduced. When there are $K$ non-zero coefficients, our algorithm recovers the Möbius transform in $O(Kn)$ samples and $O(Kn^2)$ time asymptotically under certain assumptions, the first non-adaptive algorithm to do so. We also uncover a surprising connection between group testing and the Möbius transform. For functions where all interactions involve at most $t$ inputs, we use group testing results to compute the Möbius transform with $O(Kt \log n)$ sample complexity and $O(K \operatorname{poly}(n))$ time. A robust version of this algorithm withstands noise and maintains this complexity. This marks the first $n$ sub-linear query complexity, noise-tolerant algorithm for the Möbius transform. In several examples, we observe that representations generated via sparse Möbius transform are up to twice as faithful to the original function, as compared to Shapley and Banzaf values, while using the same number of terms.

## 1 Introduction

As machine learning models become increasingly complex, our ability to interpret them has not kept pace. A natural question to ask is: What is the most fundamental interpretable representation of the functions we learn? The Shapley value [1], a concept from cooperative game theory, has become a popular way to interpret model predictions [2] by assigning importance scores to individual inputs such as features, data samples or tokens. This value represents the weighted average marginal contribution of an input, quantifying the change in the function's output when that input is included. Recent research has expanded the scope of interpretability to encompass sets of inputs [3, 4], capturing the collective influence of input combinations and their synergies on model predictions. Central to this advancement is the Möbius Transform [5], a mathematical transformation that projects functions onto a fundamental interpretable basis known in game theory as the *unanimity function basis*.

The Möbius transform has a more powerful and nuanced explanation capability than the Shapley value. Consider a sentiment analysis model (BERT [6] fine-tuned on the IMDB dataset [7]) explained

38th Conference on Neural Information Processing Systems (NeurIPS 2024).

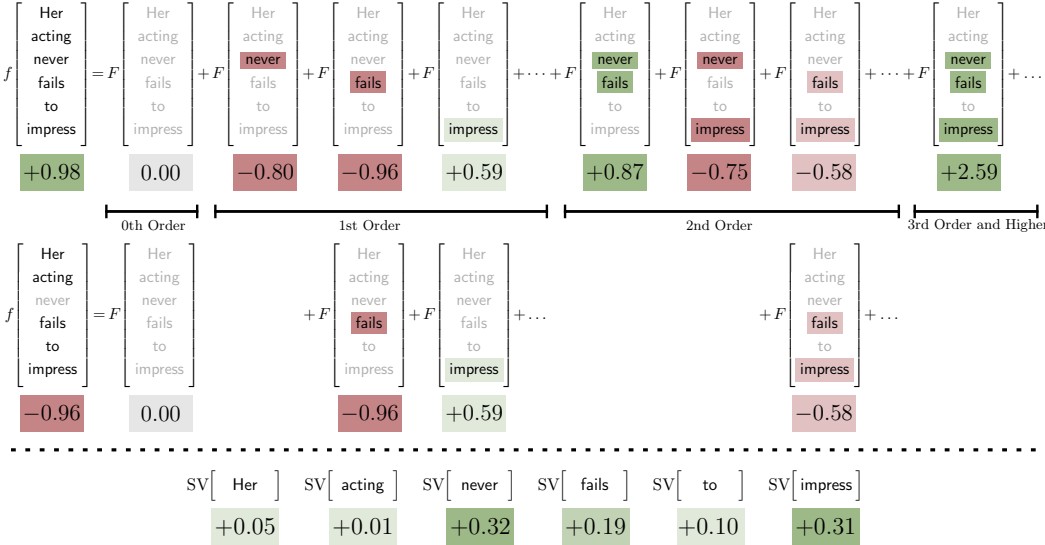

Figure 1: The movie review "Her acting never fails to impress" is passed into a BERT language model fine-tuned to do sentiment analysis [8]. Presented are $1^{st}$, $2^{nd}$ and $3^{rd}$ order Möbius coefficients, with positive interactions in green and negative in red computed via (1). The coefficients explain how groups of words influence BERT's perception of sentiment. For instance, while *never* and *fails* have strong negative sentiments individually, when combined, they impose a profound positive sentiment. In the second row, the word *never* is deleted, resulting in a large change in sentiment. In contrast, the Shapley values of each word $\text{SV}(\cdot)$, presented at the bottom of the figure, are less informative.

using both Shapley values and the Möbius transform as depicted in Fig. 1. The model's objective is to classify the sentiment of the review as positive or negative. The Möbius transform assigns a score to all word subsets within a sentence. For instance, in the sentence "Her acting never fails to impress" each subset of words is evaluated—positive interactions receive positive scores, and negative interactions, negative scores. Summing these scores yields the overall sentiment $+0.98$. This granular analysis reveals the model's understanding of linguistic constructs like double negatives, as seen in the interaction between *never* and *fails*, and the inherent positivity of words like *impress*. When the word *never* is masked, interactions involving never are excluded, shifting the sentiment negatively to $-0.96$.

This level of detail is not readily available with the Shapley value, which assigns scores to individual words without considering their interplay. The value of the Möbius transform is apparent, but given its complex structure, is it possible to compute efficiently?

In general, to compute a Möbius transform over $n$ features requires $2^n$ inferences (masking over all $2^n$ subsets of features), as well as $n2^n$ time using a divide-and-conquer approach similar to that of the Fast Fourier Transform (FFT) algorithm. GPT-4 currently supports in the range of 8000 words-per-prompt, and context length will continue to grow with new architectures [9]. Running inference $2^{8000}$ times is not even close to possible, and even if you could, $2^{8000}$ coefficients are hardly interpretable! In Fig. 1 we see that many coefficients are *insignificant*. This is typical. The solution to the computational problem is to just focus on computing the largest Möbius interactions and ignore the small ones. Is this possible in a systematic way? Yes—assuming that only $K$ Möbius coefficients (which $K$ values are significant is unknown) are non-zero, our algorithm enables us to intelligently query the model to significantly reduce the number of samples of that are required to $O(Kn)$ with $O(Kn^2)$ time. We also explore the regime where the non-zero interactions occur between at most $t$ inputs, with $t \ll n$, showing that only $O(Kt\log(n))$ samples are required in $O(K\,\text{poly}(n))$ time. We also have a robust algorithm that allows for some noise in the sampling process, effectively relaxing the constraint that the insignificant coefficients are exactly zero while maintaining the same complexities.

**Defining the Möbius Transform**  We define a value function for a model with $n$ inputs across subsets $S \subseteq [n]$ denoted as $f(S)$. The construction of this function varies based on the model: in Fig. 1, words not in $S$ might be masked or omitted. In other cases, we might take a conditional expectation over words not in $S$. To facilitate later discussion on group testing, we express the function as $f : \mathbb{Z}_2^n \to \mathbb{R}$, where $f(S) = f(\mathbf{m})$ with $S = \{i : m_i = 1\}$. The relationship between $f : \mathbb{Z}_2^n \to \mathbb{R}$

and its Möbius transform $F : \mathbb{Z}_2^n \rightarrow \mathbb{R}$ is characterized by the forward and inverse transforms:

$$\text{Inverse:} \quad f(\mathbf{m}) = \sum_{\mathbf{k} \leq \mathbf{m}} F(\mathbf{k}), \qquad \text{Forward:} \quad F(\mathbf{k}) = \sum_{\mathbf{m} \leq \mathbf{k}} (-1)^{\mathbf{1}^{\mathrm{T}}(\mathbf{k}-\mathbf{m})} f(\mathbf{m}), \qquad (1)$$

where $\mathbf{k} \leq \mathbf{m}$ means that $k_i \leq m_i \ \forall i$. This transform acts as a bridge, connecting various importance metrics, which can be expressed as projections onto a subset of the Möbius basis. The Shapley value $\mathrm{SV}(i)$ and Banzhaf value $\mathrm{BZ}(i)$ for feature $i$ is elegantly represented within this framework:

$$\mathrm{SV}(i) = \sum_{\mathbf{k}:k_i=1} \frac{1}{|\mathbf{k}|} F(\mathbf{k}), \qquad \mathrm{BZ}(i) = \sum_{\mathbf{k}:k_i=1} \frac{1}{2^{|\mathbf{k}|-1}} F(\mathbf{k}). \qquad (2)$$

These relationships are foundational to the definition of the Shapley and Bhanzaf values themselves. The left equality appears as Eq. 10 in Lloyd Shapley's original report from 1952 [1] where the concept of Shapley value was first introduced and is central to his derivation of a closed-form expression.

## 1.1 Related Works and Applications

This work is inspired by the literature on sparse Fourier transforms, which began with [10, 11, 12]. The sparse Boolean Fourier (Hadamard) transform [13, 14] is most relevant.

**Group Testing** This manuscript establishes a strong connection between the interaction identification problem and group testing [15]. Group testing was first described by Dorfman [16], who noted that when testing soldiers for syphilis, pooling blood samples from many soldiers, and testing the pooled blood samples reduced the total number of tests needed. [17] is the first work to exploit group testing in a feature selection/importance problem, using a group testing matrix in their algorithm. [18] also mentions group testing in relation to Shapley values.

**Möbius Transform** Möbius transforms [5] have been studied in the pseudo-Boolean (set) function literature, and dates back to at least [19]. [20] develops a framework for computing sparse transforms of pseudo-Boolean functions. They do not directly consider the Möbius transform as we define it, but one can apply their algorithm to compute a $K$ sparse transform in $O(nK)$ *adaptive* samples and $O(K^2 n)$ time. In the sparse and noiseless setting, our algorithm improves on this by being fully non-adaptive and having lower time complexity in most non-trivial settings. [20] does not consider the important low degree setting and does not consider robustness to noise (approximate sparsity), which are critical aspects of this work. In [21], the authors show that a classifier satisfying certain properties can be well represented by a sparse and low degree Möbius transform.

**Explainability** [2] proposes model explanation via pseudo-Boolean functions approximated by Shapley values, effectively utilizing only first-order Möbius coefficients. Constructing these functions, [22, 23, 24, 25] especially for generative models with complex outputs [26, 27, 28], is an ongoing research area. [3] presents the Taylor-Shapley interaction index (STII), scoring interactions up to size $t$. For sets smaller than $t$, STII are exactly Möbius coefficients. [4] introduces the Faithful Shapley Interaction index (FSI), which computes scores via projection onto up to $t^{th}$ order Möbius coefficients. [29] develops methods for computing FSI, STII, and other interaction indices. The relationship between the Möbius transform, FSI, STII, Shapley value, and Banzhaf value is detailed in Appendix A.

**Data Valuation** In data valuation [18] the goal is to assign an importance score to data, either to determine a fair price [30] or to curate a more efficient dataset [31]. A feature of this problem is the high cost of getting a sample since we need to determine the accuracy of our model when trained on different subsets of data, making sample complexity of critical importance. [32, 33] try to approximate this by looking at the accuracy of partially trained models, though this introduces sampling noise.

## 1.2 Main Contributions

Our algorithm and proofs are deeply *interdisciplinary*, and the contributions of this paper are theoretical. We use modern ideas spanning across signal processing, algebra, coding and information theory, and group testing to address the important problem of interpretability at the forefront of machine learning. The main contributions of this manuscript are:

- For a function with $K$ non-zero Möbius coefficients chosen uniformly at random, the Sparse Möbius Transform (SMT) algorithm exactly recovers the transform $F$ in $O(Kn)$ samples and $O(Kn^2)$ time in the limit as $n \rightarrow \infty$ with $K$ growing at most as $2^{n\delta}$ with $\delta \leq \frac{1}{3}$.

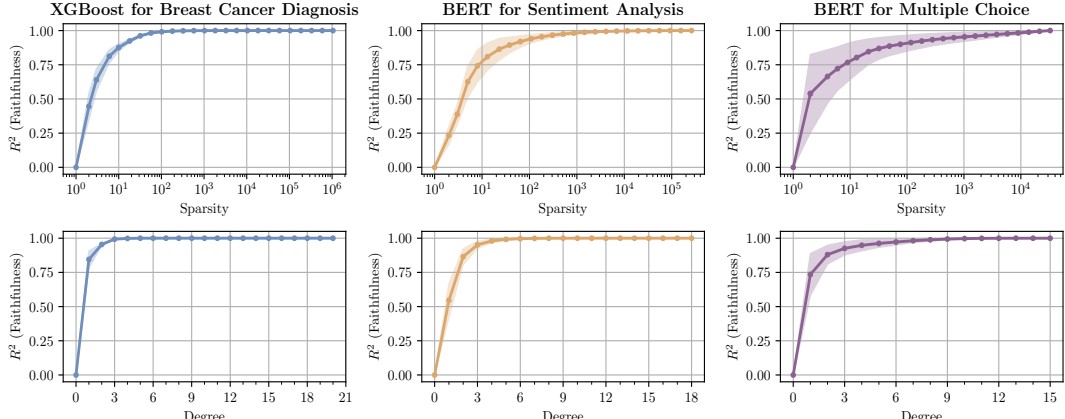

Figure 2: These plots are strong indicators that sparsity and low degree assumptions are worthy of consideration. We consider three different learning tasks. The left-most plot shows results from an XGBoost [34] model used for breast cancer diagnosis. The middle plot shows results from word-level sentiment analysis task using a BERT model [8] like in Fig. 1. The right-most plot shows results from a multiple choice question and answer task also using a BERT model [35]. Error bars represent standard deviation over 10 different instances. Details for each setting are in Appendix B. In all cases, the number of features $n \approx 20$, for which it is possible to perform the full Möbius transform. On the top row, we plot achievable faithfulness $R^2$ as a function of sparsity. We observe that in *all* cases, faithfulness approaching 1 requires only a few thousand Möbius coefficients, motivating our sparsity assumption. The bottom row of plots considers achievable faithfulness vs. degree, i.e., what $R^2$ can be achieved using only Möbius coefficients $\hat{F}$ up to a given degree. Here we observe that in nearly all cases, low degree coefficients suffice to get quite small $R^2$, motivating our low degree assumption.

- We develop a formal connection with *group testing* and present a variant of SMT that works when all non-zero interactions are low order. If the maximum order of interaction is $t = \Theta(n^\alpha)$ where $\alpha < 0.409$ then we can compute the Möbius transform in $O(Kt \log(n))$ samples in $O(K \operatorname{poly}(n))$ time with error going to zero as $n \to \infty$ with growing $K$.
- Using robust group testing, we develop an algorithm that, under certain assumptions, computes the Möbius transform in $O(Kt \log(n))$ samples, with vanishing error as $n \to \infty$ with growing $K$.

In addition to our asymptotic analysis, we provide synthetic and real-world experiments that verify that our algorithm performs well even in the finite $n$ regime. Furthermore, our results are *non-adaptive* meaning that all samples can be computed in parallel. Code has been made publicly available [1].

**Notation**    Lowercase boldface $\mathbf{x}$ and uppercase boldface $\mathbf{X}$ denote vectors and matrices respectively. $\mathbf{x} \geq \mathbf{y}$ means that $x_i \geq y_i \ \forall i$. Multiplication is always standard real field multiplication, but **addition between two elements in $\mathbb{Z}_2$ should be interpreted as a logical OR $\vee$**. We define subtraction, of $\mathbf{x} - \mathbf{y}$ for $\mathbf{x} \geq \mathbf{y}$ by standard real field subtraction. $\bar{\mathbf{x}}$ corresponds to bit-wise negation for Boolean $\mathbf{x}$, and $\mathbf{x} \odot \mathbf{y}$ represents an element-wise multiplication.

## 2    Understanding Assumptions: Sparsity and Low Degree

Computing the forward transform (1) typically requires sampling all $2^n$ input combinations, an infeasible task, even for modest $n$. For an arbitrary $f$, one cannot do any better. In fact, the same is true of the Shapley value, yet, computational tools like SHAP [2] exist because practical functions of interest are *not arbitrary*. To help understand this, we define *faithfulness* for an explanation model $\hat{f}$:

$$R^2 = 1 - \|\hat{f} - f\|^2 / \|f\|^2 \, , \text{where } \|f\|^2 = \sum_{\mathbf{m} \in \mathbb{Z}_2^n} f(\mathbf{m})^2. \tag{3}$$

Note that this corresponds to the standard definition of $R^2$ in statistics when $f$ is zero-mean, and we generally define $f$ such that this is the case. A good explanation model should have a high $R^2$,

[1]https://github.com/basics-lab/sparseMobiusTransform

a succinct representation, and most importantly, be easily computed. For the Möbius transform, we aim to learn coefficients $\hat{F}(\mathbf{k})$ efficiently and construct $\hat{f}$ using the inverse transform (1). With no restrictions on $\hat{F}(\mathbf{k})$ we can achieve $R^2 = 1$, but this fails to meet our simplicity criterion. Fortunately, many real-world functions are *sparse*—only a few $\hat{F}(\mathbf{k})$ coefficients need to be non-zero to yield $R^2 \approx 1$. Fig. 2 considers three machine learning models for breast cancer diagnosis, sentiment analysis, and question answering respectively. In all three cases, we find that we only need a small number of Möbius coefficients to achieve $R^2 \approx 1$. Furthermore, real-world functions are *low degree*, such that those small number of non-zero coefficients satisfy $|\mathbf{k}| \leq t$ for some small $t$. This results in a much more compact representation and as we shall see, also enables efficient computation.

Fig. 2 validates our assumption for the deep-learning models mentioned above. Prior research [36, 21], have presented empirical and theoretical evidence that sparsity and low degree properties are common in well-trained models. Further investigation of the spectral properties of explanation functions could be a promising avenue for future research. Our formal statements of assumptions are given below:

**Assumption 2.1.** ($K$ Uniform Interactions) $f : \mathbb{Z}_2^n \mapsto \mathbb{R}$ has a Möbius transform of the following form: $\mathbf{k}_1, \ldots, \mathbf{k}_K$ are sampled uniformly at random from $\mathbb{Z}_2^n$, and have $F(\mathbf{k}_i) \neq 0$, $\forall i \in [K]$, but $F(\mathbf{k}) = 0$ for all other $\mathbf{k} \in \mathbb{Z}_2^n$.

**Assumption 2.2.** ($K$ $t$-Degree Interactions) $f : \mathbb{Z}_2^n \mapsto \mathbb{R}$ has a Möbius transform of the following form: $\mathbf{k}_1, \ldots, \mathbf{k}_K$ are sampled uniformly from $\{\mathbf{k} : |\mathbf{k}| \leq t, \mathbf{k} \in \mathbb{Z}_2^n\}$, and have $F(\mathbf{k}_i) \neq 0$, $\forall i \in [K]$, but $F(\mathbf{k}) = 0$ other $\mathbf{k} \in \mathbb{Z}_2^n$.

**Assumption Limitations** By assuming that the non-zero coefficients are uncorrelated and uniformly distributed, we aim to understand the *fundamental difficulty* in learning a sparse Möbius transform. Correlation between non-zero coefficients means identifying one coefficient would tell us information about the locations of the others, which can be further exploited. The existence of a scheme that works well under the uniform setting suggests that it is possible to solve the problem where correlations between interactions exist. We also consider *exact* sparsity in our assumptions. In practice, these "zero" coefficients may instead have some small magnitude. We investigate this in Section 4.

## 3 Algorithm Overview

### 3.1 Subsampling and Aliasing

First we perform functional *subsampling*: For some $b < n$ we construct $u$ according to

$$u(\boldsymbol{\ell}) = f(\mathbf{m}_{\boldsymbol{\ell}}), \quad \boldsymbol{\ell} \in \mathbb{Z}_2^b, \quad \mathbf{m}_{\boldsymbol{\ell}} \in \mathbb{Z}_2^n, \quad (4)$$

where we have the freedom to choose $\mathbf{m}_{\boldsymbol{\ell}}$. Critically, the Möbius transform of $u$, denoted $U$, is related to $F$ via the well-known signal processing phenomenon of *aliasing*:

$$U(\mathbf{j}) = \sum_{\mathbf{k} \in \mathcal{A}(\mathbf{j})} F(\mathbf{k}), \quad (5)$$

where $\mathcal{A}(\mathbf{j})$ corresponds to an *aliasing set* determined by $\mathbf{m}_{\boldsymbol{\ell}}$. Fig. 3 shows this subsampling procedure on a "sparsified" version of our sentiment analysis example using different $\mathbf{m}_{\boldsymbol{\ell}}$. Our goal is to choose $\mathbf{m}_{\boldsymbol{\ell}}$ such that the non-zero values of $F(\mathbf{k})$ are uniformly spread across the aliasing sets, since that makes them easier to recover. If only a single $\mathbf{k}$ with non-zero $F(\mathbf{k})$ ends up in an aliasing set $\mathcal{A}(\mathbf{j})$, we call it a *singleton*. In Fig. 3, our

**Algorithm 1** Sparse Möbius Transform (SMT)

1: **Input:** $\mathbf{H}_c \in \mathbb{Z}_2^{b \times n}$ for $c = 1, \ldots, C$
2: $\qquad \mathbf{D}_c \in \mathbb{Z}_2^{P \times n}$ for $c = 1, \ldots, C$
3: $\hat{F}(\mathbf{k}) \leftarrow 0 \ \forall \mathbf{k}; \ \mathcal{K} \leftarrow \emptyset;$
4: **for** $c = 1$ **to** $C$ **do**
5: $\quad$ **for** $p = 1$ **to** $P$ **do**
6: $\qquad u_{c,p}(\boldsymbol{\ell}) \leftarrow f\left(\overline{\mathbf{H}_c^{\mathsf{T}} \overline{\boldsymbol{\ell}} + \mathbf{d}_{c,p}}\right), \forall \boldsymbol{\ell} \in \mathbb{Z}_2^b$
7: $\qquad U_{c,p} \leftarrow \text{FastMobius}(u_{c,p})$
8: $\quad$ **end for**
9: **end for**
10: $\mathcal{S} = \{(c, \mathbf{j}, \mathbf{k}, v) : \text{Detect}(\mathbf{U}_c(\mathbf{j})) = \mathcal{H}_S(\mathbf{k}, v)\}$
11: **while** $|\mathcal{S}| > 0$ **do**
12: $\quad$ **for** $(c, \mathbf{j}, \mathbf{k}, v) \in \mathcal{S}$ with $\mathbf{k} \in \mathcal{K}$ **do**
13: $\qquad \hat{F}(\mathbf{k}) \leftarrow v; \mathcal{K} \leftarrow \mathcal{K} \cup \{\mathbf{k}\}$
14: $\qquad$ **for** $c' = 1$ **to** $C$ **do**
15: $\qquad\quad \text{res} \leftarrow \mathbf{U}_{c'}(\mathbf{H}_{c'}\mathbf{k}) - \hat{F}(\mathbf{k})(1 - \mathbf{D}_{c'}\mathbf{k})$
16: $\qquad\quad \mathbf{U}_{c'}(\mathbf{H}_{c'}\mathbf{k}) \leftarrow \text{res}$
17: $\qquad$ **end for**
18: $\quad$ **end for**
19: $\quad$ Update $\mathcal{S}$ : Re-run $\text{Detect}(\cdot)$
20: **end while**
21: **Output:** $\hat{F}$

first subsampling generated two singletons, while our second one generated only one. Maximizing the number of singletons is one of our goals since we can ultimately use those singletons to construct the Möbius transform. In this work, we have determined two different subsampling procedures that perform well under our two assumptions:

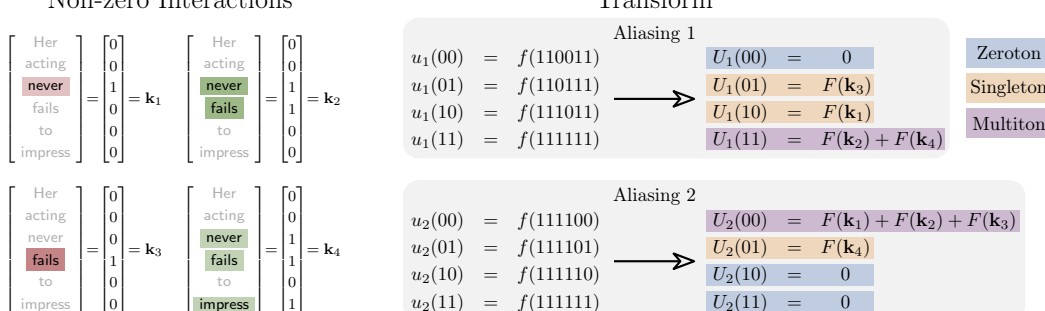

Figure 3: This figure considers a "sparsified" version of the Möbius coefficients depicted in Fig 1, keeping only the largest 4 depicted. Two different sampling choices are shown, as well as the resulting aliasing sets. In the first aliasing set, there is one zeroton, two singletons, and one multiton. In the second aliasing set, there are two zerotons, one singleton, and one multiton.

**Lemma 3.1.** *Choose* $\mathbf{m}_\ell = \overline{\mathbf{H}^T \ell}$, *which results in* $\mathcal{A}(\mathbf{j}) = \{\mathbf{k} : \mathbf{H}\mathbf{k} = \mathbf{j}\}$. $\mathbf{H}$ *is chosen as follows:*

1. *Under Assumption 2.1, we choose* $\mathbf{H} = [\mathbf{I}_{b \times b} \mathbf{0}_{b, n-b}]$, *or any column permutation of this matrix.*

2. *Under Assumption 2.2 with* $t = \Theta(n^\alpha)$ *for some* $\alpha \leq 0.409$, *there exists a matrix* $\mathbf{H}$ *chosen from* $b$ *rows of a near constant column weight group testing matrix.*

*With the given* $\mathbf{H}$, *non-zero indices are mapped to the* $2^b$ *sampling sets* $\mathcal{A}(\mathbf{j})$ *independently and uniformly at random asymptotically, thus maximizing the number of singletons when* $b = \Theta(\log(K))$.

In both cases the matrix $\mathbf{H}$ can be viewed as a part of a *group testing matrix*. This is explicit under Assumption 2.2, but under Assumption 2.1, $\mathbf{H}$ can be viewed as part of an individual testing matrix, where elements are tested one-by-one. These matrices are optimal in an information-theoretic sense because they *achieve an optimal rate* asymptotically in their respective setting. We can think of rate as the amount of information about $\mathbf{k}$ what we get from the matrix product $\mathbf{H}\mathbf{k}$ (see [15] for a formal definition). For example, if the matrix $\mathbf{H}\mathbf{k}$ was always the same for all $\mathbf{k}$, then we would get little information about $\mathbf{k}$ from $\mathbf{H}\mathbf{k}$, and we would say $\mathbf{H}$ has a *low rate*. Conversely, if $\mathbf{H}\mathbf{k}$ is different for different $\mathbf{k}$ then the product $\mathbf{H}\mathbf{k}$ provides more information about $\mathbf{k}$. Information theoretically, maximizing the rate of $\mathbf{H}$ can be thought of as maximizing the *entropy* [37] of $\mathbf{H}\mathbf{k}$. It is this connection between rate and the randomness of the product $\mathbf{H}\mathbf{k}$ that enables us to prove Lemma 3.1.

A detailed discussion of Lemma 3.1 is in Appendix C.2, with an enhanced version for independence across multiple $\mathbf{H}$, as is required for our overall result. The proof of this lemma touches many areas of mathematics, including the theory of monoids, information theory, and optimal group testing.

## 3.2 Singleton Detection and Identification

Singletons are useful, but we cannot immediately use them to recover $F(\mathbf{k})$. We first need to know that a given $U(\mathbf{j})$ *is a singleton*. Secondly, we need to identify the value of $\mathbf{k}$ that singleton corresponds to. Section 4 discusses both tasks. For now, we discuss the rest of the algorithm assuming that we can accomplish both tasks.

## 3.3 Message Passing to Resolve Collisions

Since we don't know the non-zero indices beforehand, collisions between multiple non-zero indices in the same aliasing set are inevitable. These are called *multitons*. One approach to deal with these multitons is to repeat the procedure over again:

$$u_c(\ell) = f(\mathbf{m}_{c,\ell}) \iff U_c(\mathbf{j}) = \sum_{\mathbf{k} \in \mathcal{A}_c(\mathbf{j})} F(\mathbf{k}),$$

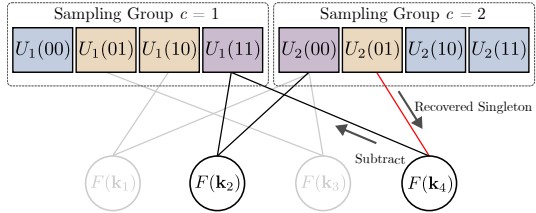

Figure 4: Depiction of our peeling message passing algorithm for the samples in Fig. 3. The singleton in $U_2(01)$ is subtracted (peeled) so we can resolve $F(\mathbf{k}_2)$ from $U_1(11)$.

$c = 1, \ldots, C$. Each time, we get different aliasing sets $\mathcal{A}_c(\mathbf{j})$ resulting in different singletons, and thus find different $\mathbf{k}$ with non-zero $F(\mathbf{k})$. While this approach works, a better approach is to combine this idea with *message passing* to use known non-zero indices and values $(\mathbf{k}, F(\mathbf{k}))$ to resolve these multitons and turn them into singletons. The type of message passing algorithm we use is called *graph peeling*. The aliasing structure can be represented as a bipartite graph. Each $U_c(\mathbf{j})$ is a *check node*, and each non-zero coefficient $F(\mathbf{k})$ is a *variable node*. The variable node $F(\mathbf{k})$ is connected to the check node $U_c(\mathbf{j})$ if $\mathbf{H}_c\mathbf{k} = \mathbf{j}$. Fig. 4 constructs this bipartite graph for the aliasing in Fig. 3. Note that $U_1(11) = F(\mathbf{k}_2) + F(\mathbf{k}_4)$ is a multiton; however, in the other sub-samping group $U_2(01) = F(\mathbf{k}_4)$ is a singleton. Once we resolve $U_2(01)$, we can simply subtract $F(\mathbf{k}_4)$ from $U_1(11)$, allowing us to create a new singleton, and extract $F(\mathbf{k}_2)$. The remaining values of $F$ both appear as singletons in the first sampling group, so we can resolve all 4 non-zero interactions $F$ with only 8 (7 unique) samples. Peeling algorithms were popularized in information and coding theory for decoding fountain codes [38] and have since been applied to variety of applications in communications [39] and signal processing [13, 40]. They can be analyzed using density evolution theory [41], which we also use as part of our proof.

## 4 Singleton Detection and Identification

We have discussed how to subsample efficiently to maximize singletons and how to use message passing to recover as many coefficients as possible. Now we discuss (1) how to identify singletons and (2) how to determine the $\mathbf{k}^*$ corresponding to the singleton. The following result is key:

**Lemma 4.1.** *Consider $\mathbf{H} \in \mathbb{Z}_2^{b \times n}$, and $f : \mathbb{Z}_2^n \mapsto \mathbb{R}$, and some $\mathbf{d} \in \mathbb{Z}_2^n$. If $U$ is the Möbius transform of $u$, and $F$ is the Möbius transform of $f$ we have:*

$$u(\boldsymbol{\ell}) = f\left(\overline{\mathbf{H}^T\overline{\boldsymbol{\ell}} + \mathbf{d}}\right) \iff U(\mathbf{j}) = \sum_{\mathbf{k} \leq \overline{\mathbf{d}} \text{ s.t. } \mathbf{H}\mathbf{k}=\mathbf{j}} F(\mathbf{k}). \quad (6)$$

The proof can be found in Appendix C.4. The form of (6) allows us to shrink the aliasing set in a controlled way. Define $\mathbf{d}_{c,0} := \mathbf{0}_n$, and $\mathbf{D}_c \in \mathbb{Z}_2^{P \times n}$ for some $P > 0$. The $i^{\text{th}}$ row of $\mathbf{D}_c$ is denoted $\mathbf{d}_{c,p}, p = 1 \ldots, P$. Using these vectors, we construct $C(P + 1)$ different subsampled functions $u_{c,p}$:

$$u_{c,p}(\boldsymbol{\ell}) = f\left(\overline{\mathbf{H}_c^T\overline{\boldsymbol{\ell}} + \mathbf{d}_{c,p}}\right), \ \forall \boldsymbol{\ell} \in \mathbb{Z}_2^b. \quad (7)$$

We compute the Möbius transform of each $u_{c,p}$ denoted by $U_{c,p}$ and construct a vector-valued function $\mathbf{U}_c(\mathbf{j}) := [U_{c,0}(\mathbf{j}), \ldots, U_{c,P}(\mathbf{j})]^T$. The goal of singleton detection is to identify when $\mathbf{U}_c(\mathbf{j})$ reduces to a single term, and for what value $\mathbf{k}$ that term corresponds to. To do so, we define the $\text{Type}(\cdot)$:

1. $\text{Type}(\mathbf{U}_c(\mathbf{j})) = \mathcal{H}_Z$ denotes a *zeroton*, for which there does not exist $F(\mathbf{k}) \neq 0$ s.t. $\mathbf{H}\mathbf{k} = \mathbf{j}$.
2. $\text{Type}(\mathbf{U}_c(\mathbf{j})) = \mathcal{H}_S(\mathbf{k}, F(\mathbf{k}))$ denotes a *singleton* with only one $\mathbf{k}$ with $F(\mathbf{k}) \neq 0$ s.t. $\mathbf{H}\mathbf{k} = \mathbf{j}$.
3. $\text{Type}(\mathbf{U}_c(\mathbf{j})) = \mathcal{H}_M$ denotes a *multiton* with more than one $\mathbf{k}$ with $F(\mathbf{k}) \neq 0$ s.t. $\mathbf{H}\mathbf{k} = \mathbf{j}$.

To describe our type estimation rule, we define the following ratios between elements of $\mathbf{U}_c(\mathbf{j})$:

$$y_{c,p} := 1 - \frac{U_{c,p}(\mathbf{j})}{U_{c,0}(\mathbf{j})}, \ \ p = 1, \ldots, P, \quad (8)$$

and construct the vector $\mathbf{y}_c := [y_{c,1}, \ldots, y_{c,P}]^T$. Then, our estimate for the type is given by

$$\text{Detect}(\mathbf{U}_c(\mathbf{j})) := \begin{cases} \mathcal{H}_Z, & \mathbf{U}_c(\mathbf{j}) = \mathbf{0} \\ \mathcal{H}_M, & \mathbf{y}_c \notin \{0,1\}^P \\ \mathcal{H}_S(\mathbf{k}, F(\mathbf{k})), & \mathbf{y}_c \in \{0,1\}^P. \end{cases} \quad (9)$$

By considering the definition of $\mathbf{U}_c$ it is possible to show that if $\text{Type}(\mathbf{U}_c(\mathbf{j})) = \mathcal{H}_S(\mathbf{k}^*, F(\mathbf{k}^*))$, then $\mathbf{y}_c = \mathbf{D}_c\mathbf{k}^*$. When we have a singleton taking $\mathbf{D}_c = \mathbf{I}$, and thus $P = n$ suffices to recover $\mathbf{k}^*$. So long as the non-zero coefficients, $F(\mathbf{k})$ are not chosen in an adversarial way, this choice of $\mathbf{D}_c$ also ensures that $\text{Detect}(\mathbf{U}_c(\mathbf{j})) = \text{Type}(\mathbf{U}_c(\mathbf{j}))$. For the purposes of our formal proof, we will assume that non-zero $F(\mathbf{k})$ are drawn from an absolutely continuous joint distribution. We can't do better if we don't have any extra information about $\mathbf{k}^*$, but we can if we know $|\mathbf{k}^*| \leq t$ as we show below. Going back to our example in Fig. 3, with $\mathbf{D}_c = \mathbf{I}$ we use a total of $8 \times 7 = 56$ samples as opposed to $2^6 = 64$. While this improvement is modest at this scale, for larger problems the improvement is dramatic.

**Singleton Identification in the Low Degree Setting**    Let's say we want to determine the singleton from $U_1(10)$ in Fig. 3, and we know $|\mathbf{k}^*| \leq 1$. By exploiting group testing, it is possible to design $\mathbf{D}_c$ with fewer rows, and thus fewer overall measurements:

Figure 5: We use group testing [16] to identify the singleton $\mathbf{k}_1$, which corresponds to the first-order term "never". By designing the masking patterns with the help of group testing, we can efficiently recover interactions with $O(t \log(n))$ extra measurements.

The matrix product $\mathbf{D}_c \mathbf{k}^*$ has a different output for each $|\mathbf{k}| \leq 1$. In this case, the result $\mathbf{y}$ corresponds to the binary index of the location of the 1. It requires $P = 3$, rather than the $P = 6$ for $\mathbf{D}_c = \mathbf{I}$. If all non-zero $F(\mathbf{k})$ had satisfied $|\mathbf{k}| \leq 1$, we could use this matrix for our example in Fig. 3. However, we only have $|\mathbf{k}| \leq 3$ for non-zero $F(\mathbf{k})$ in this example, so $\mathbf{D}_c$ as in (5) does not suffice. In the case of general $|\mathbf{k}| \leq t$, [42] says that for any scaling of $t$ with $n$, there exists a group testing design $\mathbf{D}_c$ with $P = O(t \log(n))$ that can recover $\mathbf{k}^*$ in the limit as $n \to \infty$ with vanishing error in $\text{poly}(n)$ time. If we also assume that $F(\mathbf{k})$ are $\text{Detect}\,(\mathbf{U}_c(\mathbf{j}))$ has vanishing error (see Appendix C.7.2).

**Extension to Noisy Setting**    We now relax the assumption that most of the coefficients are *exactly* zero. To do this, we assume each subsampled Möbius coefficient is corrupted by noise:

$$U_{c,p}(\mathbf{j}) = \sum_{\mathbf{k} \leq \overline{\mathbf{d}}_p \text{ s.t. } \mathbf{H}_c \mathbf{k} = \mathbf{j}} F(\mathbf{k}) + Z_{c,p}(\mathbf{j}), \tag{10}$$

where $Z_{c,p}(\mathbf{j}) \overset{i.i.d.}{\sim} \mathcal{N}(0, \sigma^2)$.  There are two main changes that must be made compared to the noiseless case. First, we must place an assumption on the magnitude of non-zero coefficients $|F(\mathbf{k}_i)|$, such that the signal-to-noise ratio (SNR) remains fixed. Secondly, the matrix $\mathbf{D}_c$ must be modified. It now consists of two parts: $\mathbf{D}_c = [\mathbf{D}_c^1; \mathbf{D}_c^2]$. We design $\mathbf{D}_c^2 \in \mathbb{Z}_2^{P_2 \times n}$ as a standard noise robust Bernoulli group testing matrix with $P_2 = O(t \log(n))$ tests, which suffices for *singleton identification* under any fixed SNR [43]. However, unlike the noiseless case, the samples from the rows of $\mathbf{D}_c^2$ are not enough to ensure a vanishing error for *singleton detection* in the $\text{Detect}\,(\cdot)$ procedure. To solve this, we design $\mathbf{D}_c^1 \in \mathbb{Z}_2^{P_1 \times n}$ as a Bernoulli group testing matrix with a different parameter. In Appendix C.7.4, we show this modified version of $\text{Detect}\,(\cdot)$ has vanishing error if $P_1 = O(t \log(n))$.

# 5    Results

Now that we have discussed all components of the algorithm, we present our theoretical guarantees:

**Theorem 5.1.** *(Recovery with $K$ Uniform Interactions) Let $f$ satisfy Assumption 2.1 for some $K = O(2^{n\delta})$ with $\delta \leq \frac{1}{3}$ and let the non-zero coefficients of $F$ be drawn from an absolutely continuous distribution. For $\{\mathbf{H}_c\}_{c=1}^C$ chosen as in Lemma C.3 with $b = O(\log(K))$, $C = 3$ and $\mathbf{D}_c = \mathbf{I}$, Algorithm 1 exactly computes the transform $F$ in $O(Kn)$ samples and $O(Kn^2)$ time complexity with probability at least $1 - O(1/K)$.*

**Theorem 5.2.** *(Noise-Robust Recovery with $K$ $t$-Degree Interactions) Let $f$ satisfy Assumption 2.2 for $K = O(\text{poly}(n))$ and $t = \Theta(n^\alpha)$ with $\alpha \leq 0.409$. Assume either:*

1. *The non-zero coefficients of $F$ are drawn from an arbitrary continuous distribution, or*

2. *$U_{c,p}$ is corrupted by noise as in (10) and let non-zero coefficients satisfy $|F(\mathbf{k})| = \rho$.*

*Then, for $\{\mathbf{H}_c\}_{c=1}^C$ chosen as in Lemma C.4 with $b = O(\log(K))$, $C = 3$, and $\mathbf{D}_c$ chosen as a suitable group testing matrix, Algorithm 1 exactly computes the transform $F$ in $O(Kt \log(n))$ samples and $O(K \text{poly}(n))$ time complexity with probability at least $1 - O(1/K)$ in both the noiseless case (1) and noisy case (2).*

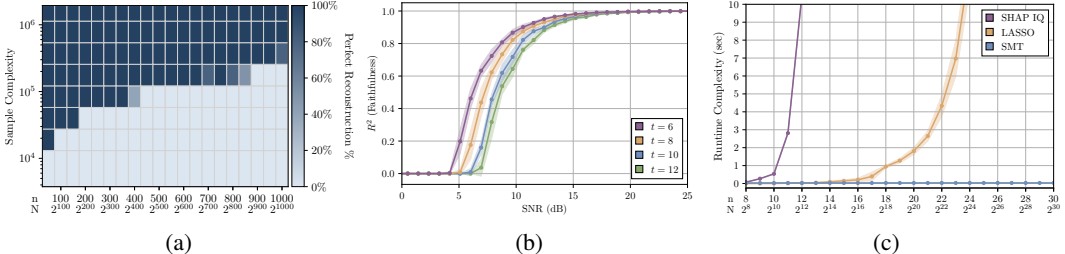

(a)                                    (b)                                    (c)

Figure 6: (a) Perfect reconstruction against $n$ and sample complexity under Assumption 2.1. Holding $C = 3$, we scale $b$ to increase the sample complexity. We observe that the number of samples required to achieve perfect reconstruction is scaling linearly in $n$ as predicted. Results are plotted across 5 runs for each choice of $b$ and $n$. (b) Plot of the noise-robust version of our algorithm. For various values of $t$, we set $n = 500$ and $K = 500$, using a group testing matrix with $P = 1000$. We plot the performance of our algorithm against SNR, measured in terms of the $R^2$. Error bands represent the standard deviation over 10 runs. (c) Runtime comparison of SMT, SHAP-IQ [29], and $t = 5$ order FSI via LASSO [4]. All are computing the Möbius transform in the setting where all non-zero interactions are order $t$, $K = 10$. SMT easily outperforms both, while the other methods quickly become intractable. Error bands represent standard deviation over 10 runs.

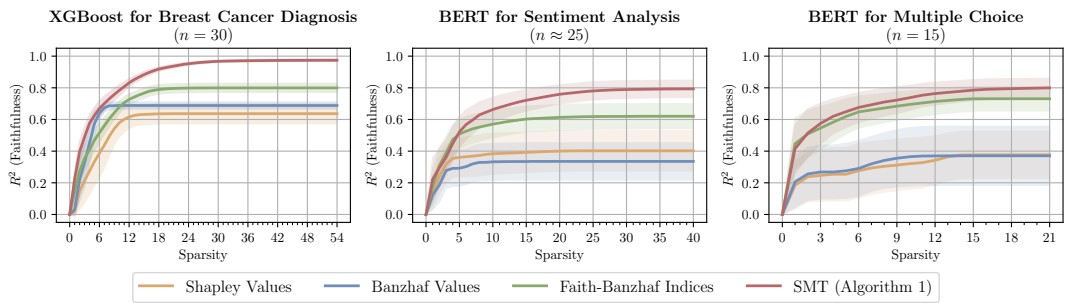

Figure 7: Since our ultimate goal is compact, meaningful and computable representations, we compare representations generated from SMT (Algorithm 1) with other popular explanation models. We plot $R^2$ (faithfulness) vs. the number of terms used in the representation (sparsity). For Shapley and Banzhaf values, to generate an $r$-sparse representation, we use the top $r$ magnitude values. For SMT and Faith-Banzhaf, we do a slightly more sophisticated refinement procedure. First order Faith-Banzhaf is included because it is the first-order metric that maximizes $R^2$. As observed in the breast cancer and sentiment analysis tasks, SMT can achieve better $R^2$ than other approaches by utilizing higher-order interactions. In the sentence-level multiple choice dataset, we observe less of a difference, since in those cases the entire answer to a question is usually contained in a single sentence, thus higher-order interactions provide little advantage. Error bands represent the standard deviation over 10 instances.

The proof of Theorem 5.1 and 5.2 is provided in Appendix C.5. It combines results from all of the parts of the algorithm we have discussed: aliasing, singleton detection and graph peeling. The requirement $|F(\mathbf{k})| = \rho$ is only due to limitations of group testing theory. In practice, we observe that a lower bound on the magnitude suffices. In addition to our theoretical results, we also conduct numerical experiments on synthetic and real word models, which are discussed below.

## 5.1 Synthetic Simulations

We tested SMT's efficacy on functions satisfying Assumption 2.1 and 2.2, setting non-zero $F(\mathbf{k})$ uniformly in $[-1, 1]$. SMT is implemented as in Algorithm 1, with group testing decoding via linear programming (see Appendix F.2). Fig. 6a is a phase transition plot that shows the percent of cases where SMT achieves $R^2 = 1$ with fixed $K = 100$ at different sample complexities and values of $n$. We *vastly outperform* the naive approach: when $n = 1000$, we get perfect reconstruction with only $10^{-294}$ percent of total samples! Furthermore, the number of samples necessary to achieve perfect reconstruction scales linearly in $n$ as predicted. Fig. 6b assesses SMT under noise for various values

of $t$, plotting $R^2$ against SNR with $K = 500$, $n = 500$, and $P = 1000$. Fig. 6c plots the runtime for SMT and competing methods. Test functions $f$ have $K = 10$ non-zero Möbius coefficients at locations that satisfy $|\mathbf{k}| = 5$ (restricted to equality due to limitations in the SHAP-IQ code at the time of running). We compare against SHAP-IQ [29] configured to compute 5[th] order FSI, as well as the method of [4] which computes 5[th] order FSI via LASSO. As shown in Appendix A, the $t$[th] order FSI are exactly the $t$[th] order Möbius coefficients for our chosen $f$. This figure exemplifies that SMT is the sole feasible method for identifying interactions on the scale of $n \geq 100$. Additional simulations and discussion can be found in Appendix E.

## 5.2   Real-World Models

Our objective is a computable, faithful, and compact representation of real-world machine-learned functions. Fig. 7 addresses this goal head-on, by plotting $R^2$ against the number of terms used in the representation (sparsity) for SMT and other popular model explanation approaches. We consider three different tasks: The first is an XGboost model for breast cancer diagnosis, and the other two are transformer-based BERT models for the tasks of sentiment analysis and multiple choice question answering respectively. Appendix B discusses the setup in great detail. For Shapley and Banzhaf values, to generate an $r$-sparse representation, we use the top $r$ magnitude values. For SMT and Faith-Banzhaf, we do a slightly more sophisticated refinement procedure using LASSO [44], described in the Appendix. We observe that for the breast cancer and sentiment analysis tasks, SMT can generate representations that, with the same number of terms, achieve a much higher $R^2$. This is done by identifying interactions between inputs that are important to the model output. Interestingly, in the case of the multiple choice model, there is less of a difference between the Faith-Banzhaf Indices and the SMT representations. This is likely because in the corresponding dataset, answers to the questions are usually contained in single sentences, making interactions less important.

## 6   Conclusion

Identifying interactions between inputs is an important open research question in machine learning, with applications to explainability, data valuation, and many other problems. We approached this problem by studying the Möbius transform, which is a representation over the fundamental interaction basis. We introduced several new tools to the problem of identifying interactions. The use of ideas from sparse signal processing and group testing has allowed SMT to operate in regimes where all other methods fail due to computational burden. Our theoretical results guarantee asymptotic exact reconstruction and are complemented by numerical simulations that show SMT performs well with finite parameters and also under a noisy model.

**Limitations**   Our assumption of independently sampled interactions was made for information-theoretic hardness and may not hold in some settings where correlated interactions exist. For instance, in the sentiment problem in Fig. 1, words with strong 2[nd] order interactions are likely to appear together in important 3[rd] order interactions. In such settings, correlation is exploitable, so a more specific algorithm can likely exploit this correlation and eliminate this assumption. Another limitation is that we have focused on taking a sparse Möbius *transform* in this work. In practice, we may be more interested in taking a sparse *projection* onto a subset of low-order terms. For unitary transforms, projection can be achieved by truncation, however, with non-orthogonal transforms like the Möbius Transform, projection is not straightforward. This is an important distinction, because there can be functions which are well-approximated by a sparse low degree Möbius projection, but do not have a sparse transform.

**Future Work**   Applying SMT to real-world tasks like understanding protein language models [45], LLM chatbots [46] or diffusion models [47] would be insightful. Working with large and complicated models will likely require further improvements to robustness—both in terms of dealing with noise from small but non-zero interactions and dealing with potential correlations between interactions. Some interesting ideas in this direction could be using more standard statistical ideas like in [29] or considering concepts from adaptive group testing. Finally, due to the connection with the Shapley or Banzhaf values (2), methods for computing the Möbius transform can also be used for computing these values by first computing the Möbius transform and then using (2).

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

# A    Relationship between Möbius Transform and Other Importance Metrics

We begin with some notation. We define the Möbius basis function (which are all possible products of inputs) as:

$$b_{\mathbf{k}}(\mathbf{m}) := \prod_{i:k_i=1} m_i. \tag{11}$$

Now we define the following sub-spaces of pseudo-Boolean function in terms of the linear span of Möbius basis functions:

$$\mathcal{M}_t := \mathrm{span}\{b_{\mathbf{k}}(\mathbf{m}) : |\mathbf{k}| \le t\}. \tag{12}$$

Now we define the projection operator $\mathrm{Proj}_\mu(f, \mathcal{D})$, as the projection of the function $f$ onto the function space $\mathcal{D}$ with respect to the measure $\mu$. If $g(\mathbf{m}) = \mathrm{Proj}_\mu(f, \mathcal{D})$, we write its decomposition as $g(\mathbf{m}) = \sum_{\mathbf{k} \in \mathbb{Z}_2^n} c(f, \mathcal{D}, \mu, \mathbf{k}) b_{\mathbf{k}}(\mathbf{m})$. Note that linear independence implies the uniqueness of this representation.

**Shapley Value**    The Shapley values $\mathrm{SV}(i)$ [1] of the inputs $m_i, i = 1, \ldots, n$ with respect to the function $f$ are [48]:

$$\mathrm{SV}(i) = c(f, \mathcal{M}_1, \sigma, \mathbf{e}_i) = \sum_{\mathbf{k}:k_i=1} \frac{1}{|\mathbf{k}|} F(\mathbf{k}), \tag{13}$$

where $\sigma$ is the Shapley kernel. $\mathrm{SV}(i) = F(\mathbf{e}_i)$ when $f$ is a linear function.

**Banzhaf Index**    The Banzhaf index $\mathrm{BZ}(i)$ of the inputs $m_i, i = 1, \ldots, n$ with respect to the function $f$ are [48]:

$$\mathrm{BZ}(i) = c(f, \mathcal{M}_1, \mu, \mathbf{e}_i) = \sum_{\mathbf{k}:k_i=1} \frac{1}{2^{|\mathbf{k}|-1}} F(\mathbf{k}), \tag{14}$$

where $\mu$ is the uniform measure. $\mathrm{BZ}(i) = F(\mathbf{e}_i)$ when $f$ is a linear function.

**Faith Shapley Interaction Index**    The $t^{\text{th}}$ order Faith Shapley interaction index $\mathrm{SV}_t(\mathbf{k})$ for $|\mathbf{k}| \le t$ [4] is

$$\mathrm{SV}_t(\mathbf{k}) = c(f, \mathcal{M}_t, \sigma, \mathbf{k}) = F(\mathbf{k}) + (-1)^{t-|\mathbf{k}|} \frac{|\mathbf{k}|}{t+|\mathbf{k}|} \binom{t}{|\mathbf{k}|} \sum_{\substack{\mathbf{p} > \mathbf{k} \\ |\mathbf{p}| > t}} F(\mathbf{p}), \tag{15}$$

where $\sigma$ is the Shapley kernel. $\mathrm{SV}_t(\mathbf{k}) = F(\mathbf{k})$ when $f$ is a $t^{\text{th}}$ order function, i.e., $F(\mathbf{k}) = 0$ when $|\mathbf{k}| > t$.

**Faith Banzhaf Interaction Index**    The $t^{\text{th}}$ order Faith Shapley interaction index $\mathrm{BZ}_t(\mathbf{k})$ for $|\mathbf{k}| \le t$ [4] is

$$\mathrm{BZ}_t(\mathbf{k}) = c(f, \mathcal{M}_t, \mu, \mathbf{k}) = F(\mathbf{k}) + (-1)^{t-|\mathbf{k}|} \sum_{\substack{\mathbf{p} > \mathbf{k} \\ |\mathbf{p}| > t}} \frac{1}{2^{|\mathbf{p}|-|\mathbf{k}|}} \binom{|\mathbf{p}| - |\mathbf{k}| - 1}{t - |\mathbf{k}|} F(\mathbf{p}), \tag{16}$$

where $\mu$ is the uniform measure. $\mathrm{BZ}_t(i) = F(\mathbf{k})$ when $f$ is a $t^{\text{th}}$ order function, i.e., $F(\mathbf{k}) = 0$ when $|\mathbf{k}| > t$.

**Shapley-Taylor Interaction Index**    The $t^{th}$ order Shapley-Taylor Interaction Index [3] $\mathrm{STII}_t(\mathbf{k})$ is:

$$\mathrm{STII}_t(\mathbf{k}) = \begin{cases} F(\mathbf{k}) & |\mathbf{k}| < t \\ c(f - f^{t-1}, \mathcal{M}_t - \mathcal{M}_{t-1}, \sigma, \mathbf{k}) & |\mathbf{k}| = t, \end{cases} \quad f^{t-1}(\mathbf{m}) = \sum_{\substack{\mathbf{k} \le \mathbf{m} \\ |\mathbf{k}| < t}} F(\mathbf{k}), \tag{17}$$

where $\sigma$ is the Shapley kernel. Explicitly, it can be shown that:

$$c(f - f^{t-1}, \mathcal{M}_t - \mathcal{M}_{t-1}, \sigma, \mathbf{k}) = \sum_{\mathbf{k} \le \mathbf{k}'} \binom{|\mathbf{k}'|}{t}^{-1} F(\mathbf{k}') \text{ for } |\mathbf{k}| = t. \tag{18}$$

As a consequence of the above, we have $\mathrm{STII}_t(\mathbf{k}) = \mathrm{SV}_t(\mathbf{k}) = F(\mathbf{k})$ when $f$ is a $t^{th}$ order function, i.e., $F(\mathbf{k}) = 0$ for $|\mathbf{k}| > t$.

## B   Experiment Details

Let $f$ be the real-world function we wish to explain. In subsections B.1, B.2, and B.3, we describe how we formed these functions for the tasks of breast cancer diagnosis, sentiment analysis, and multiple choice answering respectively. For our experiments, we plot the $R^2$ (faithfulness) for a variety of explanation models $\hat{f}$, measured through:

$$R^2 = 1 - \frac{\left\|\hat{f} - f\right\|_2^2}{\left\|f - \overline{f}\right\|_2^2}.$$

where we use the notation $\|f\|_2^2 = \sum_{\mathbf{m} \in \mathbb{Z}_2^n} f(\mathbf{m})^2$.

In Figure 2, we consider settings where $n \approx 20$, such that we can run optimization procedures to find faithful approximations that are **sparse** and **low degree**.

**Achievable Low Degree:** To find the best approximation $\hat{f}$ of up to degree $t$, we solve the following quadratic programming problem:

$$\min_{\hat{f}, \boldsymbol{\alpha}} \left\|\hat{f} - f\right\|_2^2 \tag{19}$$

$$\text{s.t.} \quad \hat{f}(\mathbf{m}) = \sum_{\mathbf{k} \leq \mathbf{m}, |\mathbf{k}| \leq t} \alpha_{\mathbf{k}}, \forall \mathbf{m}. \tag{20}$$

**Achievable Sparsity:** On the other hand, we cannot efficiently find the optimal faithful $K$-sparse approximation due to the problem's combinatorial nature. Instead, informed by the strong faithfulness of low degree approximations, we employ the following heuristic to obtain some sparse approximation.

Let $\mathcal{S}_K \subseteq \mathbb{Z}_2^n$ be a set containing the first $K$ coordinates with the lowest degree, where ties are randomly broken. With this set, we solve the following quadratic programming problem:

$$\min_{\hat{f}, \boldsymbol{\alpha}} \left\|\hat{f} - f\right\|_2^2 \tag{21}$$

$$\text{s.t.} \quad \hat{f}(\mathbf{m}) = \sum_{\mathbf{k} \leq \mathbf{m}, \mathbf{k} \in \mathcal{S}_K} \alpha_{\mathbf{k}}, \forall \mathbf{m}. \tag{22}$$

In Figure 7, we consider the four explanation models described below.

**Shapley Values:** We approximate Shapley values by iterating through permutations of the inputs [2]. For an efficient implementation of the algorithm, we use the SHAP Python package [2]. To measure the faithfulness captured by Shapley values at some sparsity level $r$, we consider approximations that only include the top-$r$ Shapley values by magnitude.

**Banzhaf Values:** We approximate Banzhaf values using the *Maximum Sample Reuse Monte Carlo* procedure described in [31]. To measure the faithfulness captured by Banzhaf values at some sparsity level $r$, we consider approximations that only include the top-$r$ Banzhaf values by magnitude.

**Faith-Banzhaf Indices:** We calculate Faith-Banzhaf indices using the regression formulation described in [4]. To measure the faithfulness captured by sparse approximations of Faith-Banzhaf indices, we modify the regression problem by adding an $\ell_1$ penalty on the values of the Faith-Banzhaf indices. We vary the penalty coefficient to obtain different levels of sparsity.

**SMT:** We run SMT (Algorithm 1) to obtain a sparse Möbius representation $\hat{F}$ with support $\text{supp}(\hat{F})$. Then, we fine-tune the values of the coefficients by solving the following regression problem over a uniformly sampled set of points $\mathcal{D} \subseteq \mathbb{Z}_2^n$:

$$\min_{\hat{f}, \boldsymbol{\alpha}} \sum_{\mathbf{m} \in \mathcal{D}} (\hat{f}(\mathbf{m}) - f(\mathbf{m}))^2$$

$$\text{s.t.} \quad \hat{f}(\mathbf{m}) = \sum_{\mathbf{k} \leq \mathbf{m}, \mathbf{k} \in \text{supp}(\hat{F})} \alpha_{\mathbf{k}}, \forall \mathbf{m}.$$

To measure the faithfulness captured by sparse approximations, we modify the regression problem by adding an $\ell_1$ penalty on the values of the Möbius coefficients. Then, we vary the penalty coefficient $\lambda$ to obtain different levels of sparsity:

$$\min_{\hat{f},\boldsymbol{\alpha}} \sum_{\mathbf{m}\in\mathcal{D}} (\hat{f}(\mathbf{m}) - f(\mathbf{m}))^2 + \lambda \sum_{\mathbf{k}\in\text{supp}(\hat{F})} |\alpha_{\mathbf{k}}|$$

$$\text{s.t.} \quad \hat{f}(\mathbf{m}) = \sum_{\mathbf{k}\leq\mathbf{m},\mathbf{k}\in\text{supp}(\hat{F})} \alpha_{\mathbf{k}}, \forall\mathbf{m}.$$

### B.1 XGBoost for Breast Cancer Diagnosis

We train an XGBoost model for classification using the Wisconsin Breast Cancer dataset [49]. This dataset contains the mean, standard deviation, and largest value of ten measurements, resulting in thirty features. For Figure 2, we use only the mean and standard deviation, resulting in twenty features. For Figure 2, we use the first ten data points in the training set and for Figure 7, we present the aggregated results over the first twenty.

To explain the XGBoost model $h$ (the probability associated with a positive classification) on each data point $x \in \mathcal{X}$, we use an interventional expected value formulation: we freeze some of the features and take an expectation over all data points by infilling the remaining features. Formally,

$$f(\mathbf{m}) = \mathbb{E}\left[h(X)|\text{do}(X_{\mathbf{m}} = x_{\mathbf{m}})\right]$$

where we use the notation $x_{\mathbf{m}} = \{x_i : m_i = 1\}$.

### B.2 BERT for Sentiment Analysis

We employ the sentiment analysis model from [8], which is built upon BERTweet [50], a RoBERTa model trained on English tweets. We take movie reviews from the IMDb Movie Reviews dataset [51]. For a particular review, we define its function as a mapping from maskings of words (using the [UNK] token) to the model's logit value associated to the correct sentiment classification.

For Figure 2, we use the first ten sentences in the dataset with 17, 18, or 19 words, where words separated through spaces in the review. Below, we include the reviews and their low degree and sparse approximations calculated with equations 20 and 22 respectively.

| | $n$ (words) | REVIEW | SENTIMENT |
|---|---|---|---|
| (a) | 18 | A rating of "1" does not begin to express how dull, depressing and relentlessly bad this movie is. | Negative |
| (b) | 18 | Hated it with all my being. Worst movie ever. Mentally- scarred. Help me. It was that bad.TRUST ME!!! | Negative |
| (c) | 19 | This is a good film. This is very funny. Yet after this film there were no good Ernest films! | Positive |
| (d) | 19 | The characters are unlikeable and the script is awful. It's a waste of the talents of Deneuve and Auteuil. | Negative |
| (e) | 18 | I don't know why I like this movie so well, but I never get tired of watching it. | Positive |
| (f) | 19 | If you like Pauly Shore, you'll love Son in Law. If you hate Pauly Shore, then, well...I liked it! | Positive |
| (g) | 17 | This is the definitive movie version of Hamlet. Branagh cuts nothing, but there are no wasted moments. | Positive |
| (h) | 19 | Without a doubt, one of Tobe Hoppor's best! Epic storytellng, great special effects, and The Spacegirl (vamp me baby!). | Positive |
| (i) | 17 | Add this little gem to your list of holiday regulars. It is sweet, funny, and endearing | Positive |
| (j) | 17 | no comment - stupid movie, acting average or worse... screenplay - no sense at all... SKIP IT! | Negative |

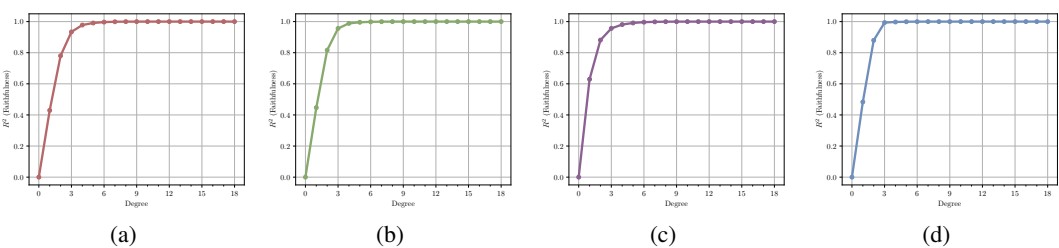

In Figure 7, we take a random sampling of reviews, with number of words spanning from 17 to 38. The reviews we used, alongside their word counts and sentiment, are included below:

### B.3 BERT for Multiple Choice

For multiple choice answering, we use a RoBERTa model [35] fine-tuned on RACE [52]: a large-scale reading comprehension dataset. This dataset contains over 28,000 passages, each containing

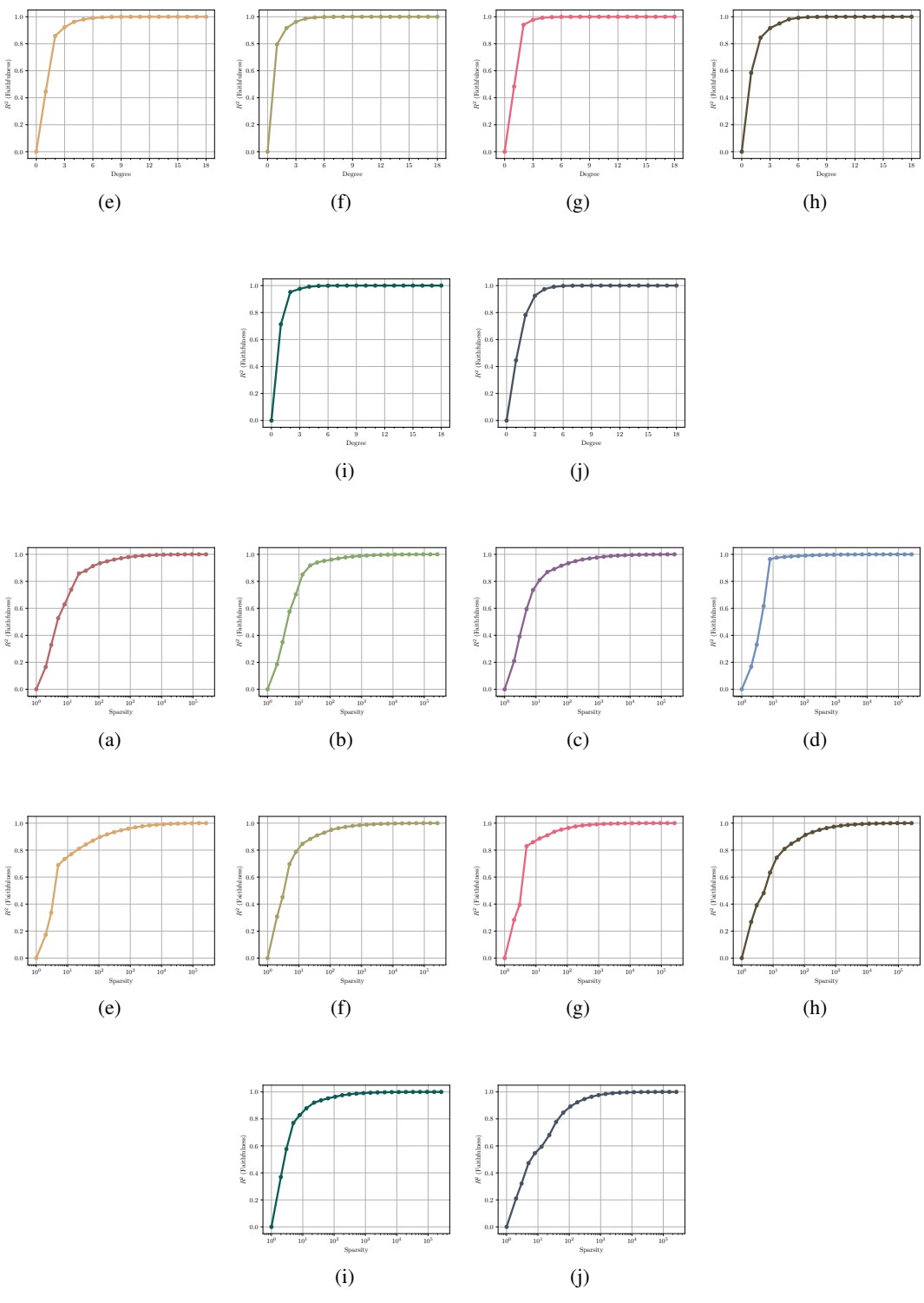

corresponding multiple-choice questions. For our experiments, we found the first ten passages with 15 sentences, and took their first multiple-choice question.

To construct the function, we consider sentence-level maskings of the passages using the [PAD] token. We pass the masked passage, alongside the multiple choice question into the RoBERTa model, and measure the logit value of the question's correct answer.

| $n$ (words) | REVIEW | SENTIMENT |
|---|---|---|
| 17 | This is the definitive movie version of Hamlet. Branagh cuts nothing, but there are no wasted moments. | Positive |
| 18 | I don't know why I like this movie so well, but I never get tired of watching it. | Positive |
| 23 | Brilliant movie. The drawings were just amazing. Too bad it ended before it begun. I´ve waited 21 years for a sequel, but nooooo!!! | Positive |
| 26 | Malcolm McDowell has not had too many good movies lately and this is no different. Especially designed for people who like Yellow filters on their movies. | Negative |
| 24 | Excellent episode movie ala Pulp Fiction. 7 days - 7 suicides. It doesnt get more depressing than this. Movie rating: 8/10 Music rating: 10/10 | Positive |
| 26 | You've got to be kidding. This movie sucked for the sci-fi fans. I would only recommend watching this only if you think Armageddon was good. | Negative |
| 27 | Despite its interesting premise, Sniper is quite tedious. With a tighter script and sharper directing it could have been electrifying; instead it plods along with little tension. | Negative |
| 29 | You may like Tim Burton's fantasies, but not in a commercial-like show off lasting 8 minutes. It demonstrates good technical points without real creativity or some established narrative pace. | Negative |
| 27 | Brilliant execution in displaying once and for all, this time in the venue of politics, of how "good intentions do actually pave the road to hell". Excellent! | Positive |
| 28 | I can't believe they got the actors and actresses of that caliber to do this movie. That's all I've got to say - the movie speaks for itself!! | Positive |
| 33 | Something does not work in this movie. There are absolutely no energies between the actors. In fact, their very acting seems frozen, sometimes amateur. Also, the script is not convincing and not reliable. | Negative |
| 24 | Great story, great music. A heartwarming love story that's beautiful to watch and delightful to listen to. Too bad there is no soundtrack CD. | Positive |
| 38 | A very carelessly written film. Poor character and idea development. The silly plot and weak acting by just about the ensemble cast didn't help. Seriously, watching this movie will NOT make you smile. It may make you retch. | Negative |
| 19 | This is a good film. This is very funny. Yet after this film there were no good Ernest films! | Positive |
| 18 | The characters are unlikeable and the script is awful. It's a waste of the talents of Deneuve and Auteuil. | Negative |

## C  Missing Proofs

### C.1  Boolean Arithmetic

Table 1 the addition and multiplication table for arithmetic between $x, y \in \mathbb{Z}_2$. We also note that $\mathbb{Z}_2$ is typically used to refer to the integer ring modulo 2. The arithmetic we are describing here is actually that of a *monoid*. Since the audience for this paper is people interested in machine learning, we continue to use $\mathbb{Z}_2$ since it is commonly used to simply refer to the set $\{0, 1\}$.

| Addition Table | | | Multiplication Table | | | Subtraction Table | | |
|---|---|---|---|---|---|---|---|---|
| $+$ | $x = 1$ | $x = 0$ | $\times$ | $x = 1$ | $x = 0$ | $-$ | $x = 1$ | $x = 0$ |
| $y = 1$ | 1 | 1 | $y = 1$ | 1 | 0 | $y = 1$ | 0 | 1 |
| $y = 0$ | 1 | 0 | $y = 0$ | 0 | 0 | $y = 0$ | N/A | 0 |

Table 1: Addition, Multiplication and Subtraction table for Boolean arithmetic in this paper. Subtraction is for $y - x$.

### C.2  Discussion of Aliasing of the Möbius Transform

When a function has many small or zero Mobius coefficients (interactions), our goal is to subsample (4) in such a way that the aliasing causes the non-zero coefficients to end up in different aliasing sets (5) (as opposed to all of them being aliased together, making them more difficult to reconstruct). Lemma C.1 is a key tool that we will use in this work to design subsampling patterns that result in good aliasing patterns.

**Lemma C.1.** *Consider* $\mathbf{H} \in \mathbb{Z}_2^{b \times n}$, $b < n$ *and* $f : \mathbb{Z}_2^n \mapsto \mathbb{R}$. *Let*

$$u(\boldsymbol{\ell}) = f\left(\overline{\mathbf{H}^T \boldsymbol{\ell}}\right), \ \forall \boldsymbol{\ell} \in \mathbb{Z}_2^b. \tag{23}$$

*If $U$ is the Mobius transform of $u$, and $F$ is the Mobius transform of $f$ we have:*

$$U(\mathbf{j}) = \sum_{\mathbf{H}\mathbf{k}=\mathbf{j}} F(\mathbf{k}). \tag{24}$$

This lemma is a powerful tool, allowing us to control the aliasing sets through the matrix $\mathbf{H}$. The proof can be found in Appendix C.3, and is straightforward, given the relationship between $u$ and

$f$. Understanding why we choose this relationship, however, is more complicated. Underlying this choice is the algebraic theory of *monoids* and abstract algebra.

As we have mentioned, our ultimate goal is to design $\mathbf{H}$ to sufficiently "spread out" the non-zero indices among the aliasing sets. Below, we define a simple and useful construction for $\mathbf{H}$.

**Definition C.2.** Consider $\{i_1, \ldots, i_b\} = I \subset [n]$, with $|I| = b$, and $\mathbf{H} \in \mathbb{Z}_2^{b \times n}$. Let $\mathbf{h}_i$ correspond to the $i^{\text{th}}$ row of $\mathbf{H}$, given by $\mathbf{h}_i = \mathbf{e}_{i_j}$, the length $n$ unit vector in coordinate $i_j$. Then if we subsample according to (23) we have:

$$U(\mathbf{j}) = \sum_{\mathbf{k} \,:\, k_i = j_i \ \forall i \in I} F(\mathbf{k}). \tag{25}$$

which happens to result in aliasing sets $\mathcal{A}(\mathbf{j}) = \{\mathbf{k} : k_i = j_i \ \forall i \in I\}$ all of equal size $2^b$. The above choice $\mathbf{H}$ actually induces a rather simple sampling procedure when we follow (23). For instance if $I = [b]$, we have:

$$u(\boldsymbol{\ell}) = f\left([\overline{\boldsymbol{\ell}}; \mathbf{1}_{n-b}]\right), \tag{26}$$

In other words, in this case, we construct samples by freezing $n - b$ of the inputs to 1 and then varying the remaining $b$ inputs across all the $2^b$ possible options. In the case where the non-zero Mobius interactions are chosen uniformly at random, this construction does a good job at spacing them out across the various aliasing sets. The following result formalizes this.

**Lemma C.3.** *(Uniform interactions) Let* $\mathbf{k}_1, \ldots, \mathbf{k}_K$ *be sampled uniformly at random from* $\mathbb{Z}_2^n$, *where* $F(\mathbf{k}_i) \neq 0, \ \forall i \in [K]$, *but* $F(\mathbf{k}) = 0$ *for all other* $\mathbf{k} \in \mathbb{Z}_2^n$. *Construct disjoint sets* $I_c \subset [n]$ *for* $c = 1, \ldots, C$, *and the corresponding matrix* $\mathbf{H}_c$ *according to Definition C.2. Let* $\mathcal{A}_c(\mathbf{j})$ *correspond to the aliasing sets after sampling with respect to matrix* $\mathbf{H}_c$. *Now define:*

$$\mathbf{j} \text{ such that } \mathbf{k}_i \in \mathcal{A}_c(\mathbf{j}) := \mathbf{j}_i^c. \tag{27}$$

*Then if* $b = O(\log(K))$, $K = O(2^{n/C})$, *in the limit as* $n \to \infty$ *with* $C = O(1)$, $\mathbf{j}_i^c$ *are mutually independent and uniformly distributed over* $\mathbb{Z}_2^b$.

The proof is given in Appendix C.6.1, and follows directly from the form of the aliasing sets $\mathcal{A}_c(\mathbf{j})$. Corollary C.3 means that using $\mathbf{H}$ as constructed in Definition C.2 ensures that we all $\mathbf{k}$ with $F(\mathbf{k}) \neq 0$ are uniformly distributed over the aliasing sets, which maximizes the number of singletons. This result, however, hinges on the fact that the non-zero coefficients are uniformly distributed. We are also interested in the case where the non-zero coefficients are all low degree. In order to induce a uniform distribution in this case, we need to exploit a *group testing* matrix.

**Lemma C.4.** *(low degree interactions) Let* $\mathbf{k}_1, \ldots, \mathbf{k}_K$ *be sampled uniformly at random from* $\{\mathbf{k} : |\mathbf{k}| \leq t, \mathbf{k} \in \mathbb{Z}_2^n\}$, *where* $F(\mathbf{k}_i) \neq 0, \ \forall i \in [K]$, *but* $F(\mathbf{k}) = 0$ *for all other* $\mathbf{k} \in \mathbb{Z}_2^n$. *By constructing* $C$ *matrices* $\mathbf{H}_c, c = 1, \ldots, C$ *from rows of a near constant column weight group testing matrix, and sampling as in (23), if* $t = \Theta(n^\alpha)$ *for* $\alpha < 0.409$, *and* $b = O(\log(K))$, $K = O(n^t)$, *in the limit as* $n \to \infty$, $\mathbf{j}_i^c$ *as defined in (27) are mutually independent and uniformly distributed over* $\mathbb{Z}_2^b$.

The proof is given in Appendix C.6.2. It relies on an information theoretic argument, exploiting a result from optimal group testing [53].

## C.3  Proof of Lemma C.1

*Proof.* Taking the Mobius transform of $u$ gives us:

$$
\begin{aligned}
U(\mathbf{k}) &= \sum_{\boldsymbol{\ell} \leq \mathbf{k}} (-1)^{\mathbb{1}^T(\mathbf{k}-\boldsymbol{\ell})} u(\boldsymbol{\ell}) \\
&= \sum_{\boldsymbol{\ell} \leq \mathbf{k}} (-1)^{\mathbb{1}^T(\mathbf{k}-\boldsymbol{\ell})} f\left( \bigodot_{i:\ell_i=0} \overline{\mathbf{h}}_i \right) \\
&= \sum_{\boldsymbol{\ell} \leq \mathbf{k}} (-1)^{\mathbb{1}^T(\mathbf{k}-\boldsymbol{\ell})} \sum_{\mathbf{r} \leq \bigodot_{i:\ell_i=0} \overline{\mathbf{h}}_i} F(\mathbf{r}) \\
&= \sum_{\boldsymbol{\ell} \in \mathbb{Z}_2^b} (-1)^{\mathbb{1}^T(\mathbf{k}-\boldsymbol{\ell})} \mathbb{1}\{\boldsymbol{\ell} \leq \mathbf{k}\} \sum_{\mathbf{r} \in \mathbb{Z}_2^n} F(\mathbf{r}) \mathbb{1}\left\{ \mathbf{r} \leq \bigodot_{i:\ell_i=0} \overline{\mathbf{h}}_i \right\} \\
&= \sum_{\mathbf{r} \in \mathbb{Z}_2^n} F(\mathbf{r}) \left( \sum_{\boldsymbol{\ell} \in \mathbb{Z}_2^b} (-1)^{\mathbb{1}^T(\mathbf{k}-\boldsymbol{\ell})} \mathbb{1}\{\boldsymbol{\ell} \leq \mathbf{k}\} \mathbb{1}\left\{ \mathbf{r} \leq \bigodot_{i:\ell_i=0} \overline{\mathbf{h}}_i \right\} \right) \\
&= \sum_{\mathbf{r} \in \mathbb{Z}_2^n} F(\mathbf{r}) I(\mathbf{r})
\end{aligned}
$$

Now let's just focus on the term in the parenthesis for now, which we have called $I(\mathbf{r})$.

**Case 1: $\mathbf{Hr} = \mathbf{k}$**

$$
I(\mathbf{r}) = \sum_{\boldsymbol{\ell} \leq \mathbf{k}} (-1)^{\mathbb{1}^T(\mathbf{k}-\boldsymbol{\ell})} \mathbb{1}\left\{ \mathbf{r} \leq \bigodot_{i:\ell_i=0} \overline{\mathbf{h}}_i \right\} \tag{28}
$$

First note that under this condition, $\boldsymbol{\ell} = \mathbf{k} \implies \mathbf{r} \leq \bigodot_{i:\ell_i=0} \overline{\mathbf{h}}_i$. To see this, note that $k_j = 0 \implies \mathbf{r} \leq \overline{\mathbf{h}}_j$. Since this holds for all $j$ such that $k_j = 0$, we have the previously mentioned implication.

Conversely, if $\ell_j < k_j$ (this means $\ell_j = 0$ AND $k_j = 1$) for some $j$, then $\mathbf{r}$ and $\mathbf{h}_j$ must overlap. Thus,

$$
\mathbb{1}\left\{ \mathbf{r} \leq \overline{\mathbf{h}}_j \right\} = 0 \implies \mathbb{1}\left\{ \mathbf{r} \leq \bigodot_{i:\ell_i=0} \overline{\mathbf{h}}_i \right\} = 0
$$

We can split $I(\mathbf{r})$ into two parts, the part where $\boldsymbol{\ell} = \mathbf{k}$ and the part where $\boldsymbol{\ell} < \mathbf{k}$:

$$
\begin{aligned}
I(\mathbf{r}) &= \mathbb{1}\left\{ \mathbf{r} \leq \bigodot_{i:k_i=0} \overline{\mathbf{h}}_i \right\} + \sum_{\boldsymbol{\ell} < \mathbf{k}} (-1)^{\mathbb{1}^T(\mathbf{k}-\boldsymbol{\ell})} \mathbb{1}\left\{ \mathbf{r} \leq \bigodot_{i:\ell_i=0} \overline{\mathbf{h}}_i \right\} \quad (\mathbf{Hr} = \mathbf{k}) & (29) \\
&= 1 + \sum_{\boldsymbol{\ell} < \mathbf{k}} 0 & (30) \\
&= 1 & (31)
\end{aligned}
$$

**Case 2: $\mathbf{Hr} \neq \mathbf{k}$**   Let $\mathbf{Hr} = \mathbf{k}' \neq \mathbf{k}$. This case itself will be broken into two parts. First let's say there is some $j$ such that $k_j = 0$ and $k_j' = 1$. Since $k_j' = 1$ we know that $\mathbb{1}\left\{ \mathbf{r} \leq \overline{\mathbf{h}}_j \right\} = 0$. Furthermore, since $\forall \boldsymbol{\ell} \in \{\boldsymbol{\ell} : \boldsymbol{\ell} \leq \mathbf{k}\}$ we have $\ell_j = 0$. Then by a similar argument to our previous one, we have $\mathbb{1}\left\{ \mathbf{r} \leq \bigodot_{i:\ell_i=0} \overline{\mathbf{h}}_i \right\} = 0 \ \forall \boldsymbol{\ell} \leq \mathbf{k}$. It follows immediately that $I(\mathbf{r}) = 0$ in this case.

Finally, we have the case where $\mathbf{k}' < \mathbf{k}$. First, if there is a coordinate $j$ such that $0 = \ell_j < k_j' = 1$, we know that $\mathbb{1}\left\{ \mathbf{r} \leq \overline{\mathbf{h}}_j \right\} = 0$ so we have $\mathbb{1}\left\{ \mathbf{r} \leq \bigodot_{i:\ell_i=0} \overline{\mathbf{h}}_i \right\} = 0 \ \forall \boldsymbol{\ell}$ s.t. $\exists j, \ell_j < k_j'$. The only $\boldsymbol{\ell}$ that remain are those such that $\mathbf{k}' \leq \boldsymbol{\ell} \leq \mathbf{k}$. It is easy to see that this is a sufficient condition for $\mathbb{1}\left\{ \mathbf{r} \leq \bigodot_{i:\ell_i=0} \overline{\mathbf{h}}_i \right\} = 1$.

$$
\begin{aligned}
I(\mathbf{r}) &= \sum_{\boldsymbol{\ell} \leq \mathbf{k}} (-1)^{\mathbb{1}^T(\mathbf{k}-\boldsymbol{\ell})} \mathbb{1}\left\{ \mathbf{r} \leq \bigodot_{i:\ell_i=0} \overline{\mathbf{h}}_i \right\} & (32) \\
&= \sum_{\mathbf{k}' \leq \boldsymbol{\ell} \leq \mathbf{k}} (-1)^{\mathbb{1}^T(\mathbf{k}-\boldsymbol{\ell})} & (33) \\
&= 0 & (34)
\end{aligned}
$$

Where the final sum is zero because exactly half of the $\ell$ have even and odd parity respectively.

Thus, the subsampling pattern becomes:

$$U(\mathbf{k}) = \sum_{\mathbf{H}\mathbf{r}=\mathbf{k}} F(\mathbf{r}).$$

$\square$

## C.4 Proof of Section 4

$$
\begin{aligned}
U(\mathbf{k}) &= \sum_{\boldsymbol{\ell}\leq\mathbf{k}}(-1)^{\mathbb{1}^T(\mathbf{k}-\boldsymbol{\ell})}u(\boldsymbol{\ell}) \\
&= \sum_{\boldsymbol{\ell}\leq\mathbf{k}}(-1)^{\mathbb{1}^T(\mathbf{k}-\boldsymbol{\ell})}f\left(\left(\bigodot_{i:\ell_i=0}\overline{\mathbf{h}}_i\right)\odot\overline{\mathbf{d}}\right) \\
&= \sum_{\boldsymbol{\ell}\leq\mathbf{k}}(-1)^{\mathbb{1}^T(\mathbf{k}-\boldsymbol{\ell})}\sum_{\mathbf{r}\leq\bigodot_{i:\ell_i=0}\overline{\mathbf{h}}_i}F(\mathbf{r})\mathbb{1}\left\{\mathbf{r}\leq\overline{\mathbf{d}}\right\} \\
&= \sum_{\boldsymbol{\ell}\in\mathbb{Z}_2^b}(-1)^{\mathbb{1}^T(\mathbf{k}-\boldsymbol{\ell})}\mathbb{1}\{\boldsymbol{\ell}\leq\mathbf{k}\}\sum_{\mathbf{r}\in\mathbb{Z}_2^n}F(\mathbf{r})\mathbb{1}\left\{\mathbf{r}\leq\bigodot_{i:\ell_i=0}\overline{\mathbf{h}}_i\right\}\mathbb{1}\left\{\mathbf{r}\leq\overline{\mathbf{d}}\right\} \\
&= \sum_{\mathbf{r}\in\mathbb{Z}_2^n}F(\mathbf{r})\mathbb{1}\left\{\mathbf{r}\leq\overline{\mathbf{d}}\right\}\left(\sum_{\boldsymbol{\ell}\in\mathbb{Z}_2^b}(-1)^{\mathbb{1}^T(\mathbf{k}-\boldsymbol{\ell})}\mathbb{1}\{\boldsymbol{\ell}\leq\mathbf{k}\}\mathbb{1}\left\{\mathbf{r}\leq\bigodot_{i:\ell_i=0}\overline{\mathbf{h}}_i\right\}\right) \\
&= \sum_{\mathbf{r}\in\mathbb{Z}_2^n}F(\mathbf{r})\mathbb{1}\left\{\mathbf{r}\leq\overline{\mathbf{d}}\right\}I(\mathbf{r}) \\
&= \sum_{\substack{\mathbf{H}\mathbf{r}=\mathbf{k}\\\mathbf{r}\leq\overline{\mathbf{d}}}}F(\mathbf{r})
\end{aligned}
$$

## C.5 Proof of Main Theorems

**Theorem 5.1.** *(Recovery with $K$ Uniform Interactions) Let $f$ satisfy Assumption 2.1 for some $K = O(2^{n\delta})$ with $\delta \leq \frac{1}{3}$ and let the non-zero coefficients of $F$ be drawn from an absolutely continuous distribution. For $\{\mathbf{H}_c\}_{c=1}^C$ chosen as in Lemma C.3 with $b = O(\log(K))$, $C = 3$ and $\mathbf{D}_c = \mathbf{I}$, Algorithm 1 exactly computes the transform $F$ in $O(Kn)$ samples and $O(Kn^2)$ time complexity with probability at least $1 - O(1/K)$.*

**Theorem 5.2.** *(Noise-Robust Recovery with $K$ $t$-Degree Interactions) Let $f$ satisfy Assumption 2.2 for $K = O(\text{poly}(n))$ and $t = \Theta(n^\alpha)$ with $\alpha \leq 0.409$. Assume either:*

1. *The non-zero coefficients of $F$ are drawn from an arbitrary continuous distribution, or*

2. *$U_{c,p}$ is corrupted by noise as in (10) and let non-zero coefficients satisfy $|F(\mathbf{k})| = \rho$.*

*Then, for $\{\mathbf{H}_c\}_{c=1}^C$ chosen as in Lemma C.4 with $b = O(\log(K))$, $C = 3$, and $\mathbf{D}_c$ chosen as a suitable group testing matrix, Algorithm 1 exactly computes the transform $F$ in $O(Kt\log(n))$ samples and $O(K\,\text{poly}(n))$ time complexity with probability at least $1 - O(1/K)$ in both the noiseless case (1) and noisy case (2).*

*Proof.* The first step for proving both Theorem 5.1 and Theorem 5.2 is to show that Algorithm 1 can successfully recover all Mobius coefficients with probability $1 - O(1/K)$ under the assumption that we have access to a $\text{Detect}\,(\mathbf{U}_c(\mathbf{j}))$ function that can output the type $\text{Type}\,(\mathbf{U}_c(\mathbf{j}))$ for any aliasing set $\mathbf{U}_c(\mathbf{j})$. Under this assumption, we use density evolution proof techniques to obtain Theorem C.5 and conclude both theorems.

Then, to remove this assumption, we need to show that we can process each aliasing set $\mathbf{U}_c(\mathbf{j})$ correctly, meaning that each bin is correctly identified as a zeroton, singleton, or multiton. Define $\mathcal{E}$ as the error

event where the detector makes a mistake in $O(K)$ peeling iterations. If the error probability satisfies $\Pr(\mathcal{E}) \le O(1/K)$, the probability of failure of the algorithm satisfies

$$\mathbb{P}_F = \Pr\left(\widehat{F} \ne F | \mathcal{E}^c\right) \Pr(\mathcal{E}^c) + \Pr\left(\widehat{F} \ne F | \mathcal{E}\right) \Pr(\mathcal{E})$$

$$\le \Pr\left(\widehat{F} \ne F | \mathcal{E}^c\right) + \Pr(\mathcal{E})$$

$$= O(1/K).$$

In the following, we describe how we achieve $\Pr(\mathcal{E}) \le O(1/K)$ under different scenarios.

In the case of uniformly distributed interactions without noise, singleton identification and detection can be performed without error as described in Section C.7.1. In the case of interactions with low degree and without noise, singleton identification and detection can be performed with vanishing error as described in Section C.7.2. Lastly, we can perform noisy singleton identification and detection with vanishing error for low degree interactions as described in Section C.7.2.

$\square$

## C.6 Density Evolution Proofs

The density evolution proof is generally separated into two parts.

- We show that with high probability, nearly all of the variable nodes will be resolved.

- We show that with high probability, the graph is a good *expander*, which ensures that if only a small number are unresolved, the remaining variable nodes will be resolved.

Whether the decoding succeeds or fails depends entirely on the graph (or rather distribution over graphs) that is induced by the algorithm. The graph ensemble is parameterized as $\mathcal{G}\left(\mathcal{D}, \{\mathbf{M}_c\}_{c \in [C]}\right)$. $\mathcal{D}$ is the support distribution. The set of non-zero Mobius coefficients $\{\mathbf{r} : \mathcal{M}[f](\mathbf{r}) \ne 0\} \sim \mathcal{D}$ is drawn from this distribution. In [13], using the arguments above it is shown that if the following conditions hold, the peeling message passing successfully resolves all variable nodes:

1. In the limit as $n \to \infty$ asymptotic check-node degree distribution from an edge perspective converges to that of independent an identically distributed Poisson distribution (shifted by 1).

2. The variable nodes have a constant degree $C \ge 3$ (This is needed for the expander property).

3. The number of check nodes $b$ in each of the $C$ sampling group is such that $2^b = O(K)$.

**Theorem C.5** ([13]). *If the above three conditions hold, the peeling decoder recovers all Mobius coefficients with probability $1 - O(1/K)$.*

In the following section, we show that for suitable choice of sampling matrix, these conditions are satisfied, both in the case of uniformly distributed and low degree Mobius coefficients.

### C.6.1 Uniform Distribution

In order to satisfy the conditions for the case of a uniform distribution of we use the matrix in Corollary C.3. We select $C = 3$ different $I_1, I_2, I_3$ such that $I_i \cap I_j = \emptyset \;\; \forall i \ne j \in \{1, 2, 3\}$. Note that this satisfies condition (2) above. Furthermore, we let $k$ scale as $O(2^{n\delta})$. In order to satisfy condition (3), we must have $\delta < \frac{1}{3}$, since each $I_i$ can consist of at most $\frac{1}{3}$ of all the coordinates.

We now introduce some notation. Let $\mathbf{g}_j(\cdot)$ represent the *hash function*, that maps a frequency $\mathbf{r}$ to a check node index $\mathbf{k}$ in each subsampling group $j = 1, \ldots, C$, i.e., $\mathbf{g}_j(\mathbf{r}) = \mathbf{H}_j \mathbf{r}$. Per our assumption, we have $K$ non-zero variable notes $\mathbf{r}^{(1)}, \ldots, \mathbf{r}^{(K)}$ chosen uniformly at random. Technically, we are sampling without replacement, however, since $\frac{K}{2^n} \to 0$, the probability of selecting a previously selected $\mathbf{r}^{(i)}$ vanishes. Going forward in this subsection, we will assume that each $\mathbf{r}_i$ is sampled with replacement for a more brief solution. A more careful analysis that deals with sampling with replacement before taking limits yields an identical result.

First, let's consider the marginal distribution of $\mathbf{g}_j(\mathbf{r}^{(i)})$ for some arbitrary $j \in [C]$ and $i \in [K]$. Assuming sampling with replacement, we have:

$$\Pr\left(\mathbf{g}_j(\mathbf{r}^{(i)}) = \mathbf{k}\right) = \Pr\left(\mathbf{r}_{I_j}^{(i)} = \mathbf{k}\right) = \prod_{m \in I_j} \Pr\left(r_m^{(i)} = k_m\right) = \frac{1}{2^b}. \tag{35}$$

Thus, we have established that the our approach induces a uniform marginal distribution over the $2^b$ check nodes. Next, we consider the independence of our bins. By assuming sampling with replacement, we can immediately factor our probability mass function.

$$\Pr\left(\bigcap_{i,j} \mathbf{g}_j(\mathbf{r}^{(i)}) = \mathbf{k}^{(i,j)}\right) = \prod_i \Pr\left(\bigcap_j \mathbf{g}_j(\mathbf{r}^{(i)}) = \mathbf{k}^{(i,j)}\right) \tag{36}$$

Furthermore, since we carefully chose the $I_i$ such that they are pairwise disjoint, we have

$$\Pr\left(\bigcap_j \mathbf{g}_j(\mathbf{r}^{(i)}) = \mathbf{k}^{(i,j)}\right) = \Pr\left(\bigcap_j \mathbf{r}_{I_j}^{(i)} = \mathbf{k}^{(i,j)}\right) = \prod_j \Pr\left(\mathbf{r}_{I_j}^{(i)} = \mathbf{k}^{(i,j)}\right) =$$
$$\prod_j \Pr\left(\mathbf{g}_j(\mathbf{r}^{(i)}) = \mathbf{k}^{(i,j)}\right), \quad (37)$$

establishing independence. Let's define an inverse load factor $\eta = \frac{2^b}{K}$. From a edge perspective, sampling with replacement with independent uniformly distributed gives us:

$$\rho_j = j\eta \binom{K}{j} \left(\frac{1}{2^b}\right)^j \left(1 - \frac{1}{2^b}\right)^{k-j}, \tag{38}$$

For fixed $\eta$, asymptotically as $K \to \infty$ this converges to:

$$\rho_j \to \frac{(1/\eta)^{j-1}e^{-1/\eta}}{(j-1)!}. \tag{39}$$

### C.6.2 Low Degree Distribution

For this proof, we take an entirely different approach to the uniform case. We instead exploit the results of Theorem F.1, which is about asymptotically exactly optimal group testing, and then make an information-theoretic argument. Let $\mathbf{X}^n$ be a group testing matrix (constructed either by an i.i.d. Bernoulli design or a constant column weight design using the parameters required for the given $n$). We don't explicitly write the dependence of $\mathbf{X}^n$ on $t$, since by invoking Theorem F.1, we assume some implicit relationship where $t = \Theta(n^\theta)$ for $\theta$ satisfying the theorem conditions. Now consider some $\mathbf{r}_n$ chosen uniformly at random from the $\binom{n}{t}$ weight $t$ binary vectors. Note that in this work we actually use what is known as the "i.i.d prior" as opposed to the "combinatorial prior" that we have just defined. As noted in [15], these are actually equivalent, so we can arbitrarily choose to work with one, and the result holds for the other as well. We define:

$$\mathbf{Y}^n = \mathbf{X}^n \mathbf{r}^n. \tag{40}$$

Furthermore, we define the decoding function $\mathrm{Dec}_n(\cdot)$, which represents the deterministic procedure that successfully recovers $\mathbf{r}$ with vanishing error probability. We have the following bounds on the entropy of $\mathbf{Y}_n$:

$$H(\mathbf{Y}^n) = H(Y_1^n) + H(Y_2^n \mid Y_1^n) + \cdots + H(Y_T^n \mid Y_1^n, \ldots, Y_{T-1}^n) \tag{41}$$
$$\leq T, \tag{42}$$

where we have used the fact that binary random variables have a maximum entropy of 1. Furthermore, by the properties of entropy we also have $H(\mathbf{Y}^n) \geq H(\mathrm{Dec}(\mathbf{Y}^n, \mathbf{X}^n) \mid \mathbf{X}^n)$. Dividing through by $T$, we have:

$$\frac{H(\mathrm{Dec}(\mathbf{Y}^n, \mathbf{X}^n) \mid \mathbf{X}^n))}{T} \leq \frac{H(\mathbf{Y}^n)}{T} \leq 1. \tag{43}$$

Let $\text{Dec}_n(\mathbf{Y}^n, \mathbf{X}^n)) = \mathbf{r}^n + \text{err}_n(\mathbf{Y}^n, \mathbf{X}^n)$. It is known (see [15]) that $\Pr(\text{err}_n(\mathbf{Y}^n, \mathbf{X}^n) \neq 0) = O(\text{poly}(T)e^{-T})$. Thus, we can bound the left-hand side as:

$$\frac{H(\text{Dec}(\mathbf{Y}^n, \mathbf{X}^n) \mid \mathbf{X}^n)}{T} = \frac{H(\mathbf{r}^n + \text{err}_n(\mathbf{Y}^n, \mathbf{X}^n) \mid \mathbf{X}^n)}{T} \tag{44}$$

$$\geq \frac{H(\mathbf{r}^n) - H(\text{err}_n(\mathbf{Y}^n, \mathbf{X}^n) \mid \mathbf{X}^n)}{T} \tag{45}$$

$$\geq \frac{H(\mathbf{r}^n) - H(\text{err}_n(\mathbf{Y}^n, \mathbf{X}^n))}{T}, \tag{46}$$

Where in (45) we have used the bound $H(A + B) \geq H(A) - H(B)$ and the fact that $\mathbf{X}^n$ and $\mathbf{r}^n$ are independent, and in (46) we have used the fact that conditioning only decreases entropy. By the continuity of entropy and Theorem F.1, we have that:

$$\lim_{n \to \infty} \frac{H(\mathbf{r}^n) - H(\text{err}_n(\mathbf{Y}^n, \mathbf{X}^n))}{T} = \lim_{n \to \infty} \frac{\log \binom{n}{t}}{T} - \lim_{n \to \infty} \frac{H(\text{err}_n(\mathbf{Y}^n, \mathbf{X}^n))}{T} = 1 - 0 = 1. \tag{47}$$

This establishes that:

$$\lim_{n \to \infty} \frac{1}{T(n)} \sum_{i=1}^{T(n)} H\left(Y_i^n \mid \mathbf{Y}_{1:(i-1)}^n\right) = 1. \tag{48}$$

Unfortunately, this does not immediately imply that *all* of the summands have a limit of 1, however, it does mean that the fraction of total summands that are less than one goes to zero (it grows as $o(T(n))$). Let $G \subset \mathbb{N}$ correspond to the set containing all the indicies $i$ of the summands that are equal to 1. By using the fact that conditioning only reduces entropy, we have

$$\lim_{n \to \infty} H\left(Y_i^n \mid \mathbf{Y}_{S_i}^n\right) = 1, \quad S_i = \{j < i, j \in G\}, \tag{49}$$

Now we define the countable sequence of random variables:

$$\bar{Y}_i = \lim_{n \to \infty} Y_i^n, \ i \in \mathbb{N}. \tag{50}$$

By continuity of entropy, and the above limit and definition, we have:

$$H\left(\bar{Y}_i \mid \bar{\mathbf{Y}}_{S_i}\right) = 1, \tag{51}$$

Noting that conditioning only decreases entropy, we immediately have that $\bar{Y}_i \sim \text{Bern}(\frac{1}{2})$. Now consider some arbitrary finite set $S^* \subset G$. We will now prove that $\{\bar{Y}_i, i \in S^*\}$ is mutually independent.

*Proof.* Let $i_1 < i_2 < \ldots < i_{|S^*|}$ be an ordered indexing of the elements of $S^*$. Furthermore, let $Q_j = \{i_q \mid 1 \leq q \leq j\}$. Assume the set $\{\bar{Y}_i, i \in Q_j\}$ is mutually independent, and use the notation $\mathbf{Y}_{Q_j}$ to represent a vector containing all of these entries. We have:

$$H(Y_{i_{j+1}}, \mathbf{Y}_{Q_j}) = H(\mathbf{Y}_{Q_j}) + H(Y_{i_{j+1}} \mid \mathbf{Y}_{Q_j}). \tag{52}$$

However, by using the fact that conditioning only decreases entropy we have:

$$1 = H(Y_{i_{j+1}} \mid \mathbf{Y}_{S_{j+1}}) \leq H(Y_{i_{j+1}} \mid \mathbf{Y}_{Q_j}) \leq H(Y_{i_{j+1}}) \leq 1, \tag{53}$$

thus,

$$H(Y_{i_{j+1}} \mid \mathbf{Y}_{Q_j}) = H(Y_{i_{j+1}}) = 1. \tag{54}$$

This leads to the following chain of implications:

$$H(Y_{i_{j+1}}, \mathbf{Y}_{Q_j}) = H(\mathbf{Y}_{Q_j}) + H(Y_{i_{j+1}}) \iff Y_{i_{j+1}} \perp\!\!\!\perp \mathbf{Y}_{Q_j}. \tag{55}$$

From this, and the initial inductive assumption, we can conclude that $\{\bar{Y}_i, i \in Q_{j+1}\}$ is mutually independent. The base case of $j = 1$ follows from the fact that a set containing just one single random variable is mutually independent. Since $Q_{|S^*|} = S^*$ the proof is complete. $\square$

Now let $L(n) = |G \cap [n]|$ we know $L = \Theta(T(n))$, which follows from the stronger result that $\lim_{n \to \infty} \frac{L(n)}{T(n)} = 1$. Take $b \leq \frac{L(n)}{C}$ By leveraging the above results, we can select our subsampling matrices $\{\mathbf{H}_i\}_{i=1}^C$ from suitable rows of $\mathbf{X}_n$. Let $S_1^{(n)}, \ldots, S_C^{(n)} \subset G \cap [n]$, $\left|S_i^{(n)}\right| = b$ and $S_i^{(n)} \cap S_j^{(n)} = \emptyset$. Then take

$$\mathbf{H}_i(n) = \mathbf{X}_{S_i,:}^n. \tag{56}$$

Due to the independence result proved above, the asymptotic degree distribution is:

$$\rho_j \to \frac{(1/\eta)^{j-1} e^{-1/\eta}}{(j-1)!}. \tag{57}$$

### C.7 Singleton Detection and Identification

In this section, we prove the main results about singleton detection and identification. For the noiseless case, we introduce another assumption:

**Assumption C.6.** (No Cancellation) Suppose the non-zero values $F(\mathbf{k}_1), \dots, F(\mathbf{k}_K)$ are sampled from a joint distribution $\mathbb{P}$ that satisfies the following condition:

$$\sum_{i \in S} F(\mathbf{k}_i) \neq 0, ; \forall S \subseteq [K], ; S \neq \emptyset. \tag{58}$$

This assumption is quite mild, and there are many classes of $\mathbb{P}$ that satisfy this assumption.

**Lemma C.7.** *Any absolutely continuous $\mathbb{P}$ satisfies Assumption C.6 a.s..*

*Proof.* For simplicity let $\mathbf{F}$ represent a $K$ dimensional random vector containing $F(\mathbf{k}_i)$ at index $i$. Let the set $\mathcal{R}(S, \alpha) = \{\mathbf{F} : \sum_{i \in S} F(\mathbf{k}_i) = \alpha\}$. Since $\mathbb{P}$ is absolutely continuous, a density $p$ exists such that:

$$\mathbb{P}(\mathbf{F} \in \mathcal{R}(S, \alpha)) = \int_{\mathcal{R}(S, \alpha)} dp. \tag{59}$$

However, $\dim(\mathcal{R}(S, \alpha)) = K - |S|$. Thus for $S \neq \emptyset$, $\mathcal{R}(S, \alpha)$ has Lebesgue measure zero, thus

$$\Pr\left(\sum_{i \in S} F(\mathbf{k}_i) = 0\right) = \mathbb{P}(\mathbf{F} \in \mathcal{R}(S, 0)) = \int_{\mathcal{R}(S,0)} dp = 0. \tag{60}$$

$\square$

#### C.7.1 Uniform Interactions Singleton Identification and Detection without Noise

For singleton identification, we observe that in the case of a singleton, $\mathbf{y}_c = \mathbf{k}^*$, thus, in the case of a singleton $\mathbf{k}^*$ can be recovered. We still need to show that the $\mathrm{Detect}$ function correctly identifies the type. We separate the proof into three parts:

(1) Prove $\mathrm{Detect}(\mathbf{U}_c(\mathbf{j})) = \mathcal{H}_Z \implies \mathrm{Type}(\mathbf{U}_c(\mathbf{j})) = \mathcal{H}_Z$.

(2) Prove $\mathrm{Detect}(\mathbf{U}_c(\mathbf{j})) = \mathcal{H}_S \implies \mathrm{Type}(\mathbf{U}_c(\mathbf{j})) = \mathcal{H}_S$.

(3) Prove $\mathrm{Detect}(\mathbf{U}_c(\mathbf{j})) = \mathcal{H}_M \implies \mathrm{Type}(\mathbf{U}_c(\mathbf{j})) = \mathcal{H}_M$.

Consider the subsampling group $\mathbf{U}_c(\mathbf{j})$ for some fixed $c, \mathbf{j}$. We first consider the case where $|\mathbf{k}|$ is not restricted, and denote the set of non-zero indices $\mathbf{k}_1, \dots, \mathbf{k}_K$ as $\mathcal{N}$.

**Proof of (1)**   Let $\mathrm{Detect}(\mathbf{U}_c(\mathbf{j})) = \mathcal{H}_Z$, and for contradiction's sake assume $\mathrm{Type}(\mathbf{U}_c(\mathbf{j})) \neq \mathcal{H}_Z$.

$$\mathrm{Detect}(\mathbf{U}_c(\mathbf{j})) = \mathcal{H}_Z \implies \sum_{\mathbf{k} \leq \bar{\mathbf{d}}_p \text{ s.t. } \mathbf{H}_c\mathbf{k}=\mathbf{j}} F(\mathbf{k}) = 0 \; \forall p, \tag{61}$$

Since $\mathrm{Type}(\mathbf{U}_c(\mathbf{j})) \neq \mathcal{H}_Z$, we have that $\mathcal{N} \cap \{\mathbf{k} : \mathbf{H}_c\mathbf{k} = \mathbf{j}\} \neq \emptyset$. Thus, if we consider the above implication for the case of $p = 0$ and noting that $\mathbf{d}_0 = \mathbf{0}$, we have:

$$\mathrm{Detect}(\mathbf{U}_c(\mathbf{j})) = \mathcal{H}_Z \implies \sum_{\mathcal{N} \cap \{\mathbf{k}:\mathbf{H}_c\mathbf{k}=\mathbf{j}\}} F(\mathbf{k}) = 0 \tag{62}$$

But considering the no cancellation condition, this is impossible, thus proving (1).

**Proof of (2)**   Note that the converse of (1) is true (the proof is immediate). Thus, proving $\mathrm{Detect}(\mathbf{U}_c(\mathbf{j})) = \mathcal{H}_S \implies \neg(\mathrm{Type}(\mathbf{U}_c(\mathbf{j})) = \mathcal{H}_M)$ is the same as proving (2). We will again use the method of contradiction, and assume $\mathrm{Detect}(\mathbf{U}_c(\mathbf{j})) = \mathcal{H}_S$ and $\mathrm{Type}(\mathbf{U}_c(\mathbf{j})) = \mathcal{H}_M$.

$$\mathrm{Detect}(\mathbf{U}_c(\mathbf{j})) = \mathcal{H}_S \implies U_{c,p}(\mathbf{j}) \in \{0, U_{c,0}(\mathbf{j})\} \; \forall p > 1. \tag{63}$$

Note that by our assumption, $U_{c,0}(\mathbf{j}) \neq 0$. By our assumption that $\mathrm{Type}(\mathbf{U}_c(\mathbf{j})) = \mathcal{H}_M$, we must have $|\mathcal{N} \cap \{\mathbf{k} : \mathbf{H}_c\mathbf{k} = \mathbf{j}\} \neq \emptyset| \geq 2$. Choose $\mathbf{k}_1, \mathbf{k}_2 \in \mathcal{N} \cap \{\mathbf{k} : \mathbf{H}_c\mathbf{k} = \mathbf{j}\}$. Given our choice of $\mathbf{d}_p$

$\exists p^* > 1$ s.t. only *one* of $\mathbf{k}_1$ or $\mathbf{k}_2 \leq \bar{\mathbf{d}}_{p^*}$. Without loss of generality we will assume $\mathbf{k}_2 \leq \bar{\mathbf{d}}_{p^*}$ and $\mathbf{k}_1 \not\leq \bar{\mathbf{d}}_{p^*}$. Now, define the following sets:

$$\mathcal{J}_0 = \mathcal{N} \cap \{\mathbf{k} : \mathbf{H}_c \mathbf{k} = \mathbf{j}\} \tag{64}$$
$$\mathcal{J}_{p^*} = \mathcal{N} \cap \{\mathbf{k} : \mathbf{H}_c \mathbf{k} = \mathbf{j} \ \mathbf{k} \leq \bar{\mathbf{d}}_{p^*}\} \tag{65}$$

We know $\mathbf{k}_2 \in \mathcal{J}_{p^*}$ and $\mathbf{k}_1 \in \mathcal{J}_0 \setminus \mathcal{J}_{p^*}$, thus $|\mathcal{J}_{p^*}| \geq 1$ and $|\mathcal{J}_0 \setminus \mathcal{J}_{p^*}| \geq 1$. With this, we can show that the implication above cannot be satisfied.

**Case 1:** $U_{c,p^*}(\mathbf{j}) = 0$.

$$U_{c,p^*}(\mathbf{j}) = 0 \implies \sum_{\mathbf{k} \in \mathcal{J}_p^*} F(\mathbf{k}) = 0. \tag{66}$$

Since $\mathcal{J}_p^*$ is not empty, from our distributional assumption, the above sum cannot be 0.

**Case 2:** $U_{c,p^*} = U_{c,0}(\mathbf{j})$.

$$U_{c,0}(\mathbf{j}) - U_{c,p^*}(\mathbf{j}) = 0 \implies \sum_{\mathbf{k} \in \mathcal{J}_0 \setminus \mathcal{J}_p^*} F(\mathbf{k}) = 0. \tag{67}$$

Since $\mathcal{J}_0 \setminus \mathcal{J}_p^*$ is not empty, from our distributional assumption, the above sum cannot be 0. This implies that $\mathrm{Type}(\mathbf{U}_c(\mathbf{j})) = \mathcal{H}_M$ must be false, thus proving (2).

**Proof of (3):** Since we have a converse for (1), a converse for (2) suffices to prove (3). The converse follows below:

$$\mathrm{Type}(\mathbf{U}_c(\mathbf{j})) = \mathcal{H}_S \implies \exists \mathbf{k}^* \text{ s.t. } U_{c,p} \in \{0, F(\mathbf{k}^*)\}, \ \forall p. \tag{68}$$

Since $F(\mathbf{k}^*) \neq 0$, we have $U_{c,0}(\mathbf{j}) \neq 0$ and all entries of $\mathbf{U}_c(\mathbf{j})$ are either $F(\mathbf{k}^*)$ or 0. Thus, $\mathrm{Detect}(\mathbf{U}_c(\mathbf{j})) = \mathcal{H}_S$.

### C.7.2 Low Degree Singleton Identification and Detection without Noise

Here we include a sketch of the argument for the noiseless part of Theorem 2, which focuses on the case $|\mathbf{k}| \leq t$ (the rest of Theorem 2 is unchanged).

(1) Prove $\mathrm{Detect}(\mathbf{U}_c(\mathbf{j})) = \mathcal{H}_Z \implies \mathrm{Type}(\mathbf{U}_c(\mathbf{j})) = \mathcal{H}_Z$.

(2+3) Prove $\mathrm{Pr}\left(\mathrm{Detect}(\mathbf{U}_c(\mathbf{j})) = \mathrm{Type}(\mathbf{U}_c(\mathbf{j}))\right) \to 1$.

**Proof of (1)** The proof is identical to above.

**Proof of (2+3)** The proof is the same, with one notable exception. In the low degree case, $p^*$ may not always exist. In this case, we can simply rely on the result of [54]. Since $\mathrm{Pr}(\hat{\mathbf{k}} \neq \mathbf{k}^*) \leq n^{-\beta}$, we correctly recover each $\mathbf{k}^*$ in the limit, meaning we must have $\mathrm{Pr}(\mathbf{D}_c \mathbf{k}_1 \neq \mathbf{D}_c \mathbf{k}_2) \to 1$ (this is equivalent to the existence of $p^*$). Thus, by the same argument this implies that $\mathrm{Detect}(\mathbf{U}_c(\mathbf{j}))$ has vanishing error in the limit. The complete argument requires us to union bound over all of the singleton identifications success, but since $T$ is linear in $\beta$, so long as $K = \mathrm{poly}(n)$, constant $\beta$ suffices for vanishing error.

### C.7.3 Singleton Identification in i.i.d. Spectral Noise

In this section, we discuss how to ensure that we can detect the true non-zero index $\mathbf{r}^*$ from the delayed samples, under the i.i.d. noise assumption. We first discuss the delay matrix itself, $\mathbf{D} \in \mathbb{Z}_2^{P_1 \times n}$. As in the noiseless case, we want to choose this matrix to be a group testing matrix. For the purposes of theory, we will choose $\mathbf{D}$ such that each element is drawn i.i.d. as a $\mathrm{Bern}\left(\frac{\nu}{t}\right)$ for some $\nu = \Theta(1)$. We denote the $i^{th}$ row of $\mathbf{D}$ as $\mathbf{d}_i$. Each group test is derived from one of the delayed samples. Under the i.i.d. spectral noise model, this means each sample has the form:

$$U_i(\mathbf{k}) = \sum_{\substack{\mathbf{Hr}=\mathbf{k} \\ \mathbf{r} \leq \bar{\mathbf{d}}_i}} F(\mathbf{r}) + Z_i(\mathbf{k}) \tag{69}$$
$$= F(\mathbf{r}^*)\mathbb{1}\left\{\mathbf{r}^* \leq \bar{\mathbf{d}}_i\right\} + Z_i(\mathbf{k}), \tag{70}$$

where $Z_i(\mathbf{k}) \sim \mathcal{N}\left(0, \sigma^2\right)$. Essentially, we can view this as a hypothesis testing problem, where we have one sample $X$, and hypothesis and the alternative are:

$$H_0 : X = Z \quad H_1 : X = F(\mathbf{r}^*) + Z, \ \ Z \sim \mathcal{N}(0, \sigma^2)$$

Furthermore, lets say the magnitude of $|F[\mathbf{k}]| = \rho$ is known. We construct a threshold test:

$$\varphi(X) = \mathbb{1}\left\{|X| > \gamma\right\} \tag{71}$$

With such a test, we can compute the cross-over probabilities:

$$p_{01} = \Pr_{H_0}\left(|X| > \gamma\right) = 2Q(\gamma/\sigma), \tag{72}$$

$$p_{10} = \Pr_{H_1}\left(|X| < \gamma\right) = \Phi((\gamma - \rho)/\sigma) - \Phi((-\gamma - \rho)/\sigma). \tag{73}$$

For the sake of simplicity, we will make the choice to choose $\gamma$ such that $p_{10} = p_{01}$. In that case, we can numerically solve for the cross-over probability which is fixed for a given signal-to-noise ratio.

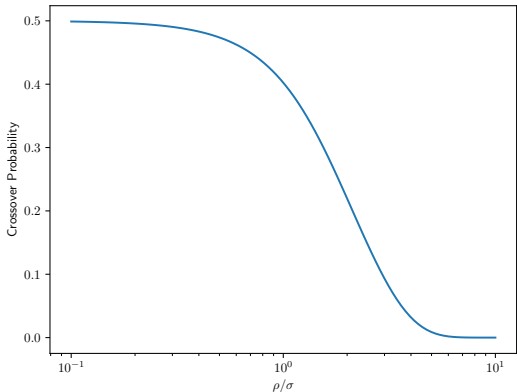

Figure 14: Symmetric cross-over probability induced by hypothesis testing problem for noisy singleton identification/detection.

Now we can use [54], to prove our desired result. let $q$ be the resulting cross-over probability for a given $\rho/\sigma$. The probability that all of our Singleton identifications succeed can computed by a union bound on $P_e = \Pr\left(\hat{\mathbf{k}} \neq \mathbf{k}\right) \leq n^{-\beta}$. If $K = \mathrm{poly}(n)$, a constant $\beta$ suffices us to drive the union bound to zero.

**Lemma C.8.** *For any fixed SNR, taking $\mathbf{D}_c$ such that each element is $\mathrm{Bern}\left(\frac{\nu}{t}\right)$, and $t = \Theta(n^\alpha)$ for $\alpha \in (0, 1)$, taking $P_1 = O(t \log(n))$ suffices to ensure that we can achieve error of bin identification failure with probability of error $O(1/K^3)$.*

### C.7.4 Singleton Detection in i.i.d. Spectral Noise

We note that the general flow of this proof follows [40], but there are several fundamental differences that make this proof overall quite different. We define $\mathcal{E}_b$ as the error event where a bin $\mathbf{k}$ is decoded wrongly, and then using a union bound over different bins and different iterations, the probability of the algorithm making a mistake in bin identification satisfies

$$\Pr(\mathcal{E}) \leq (\text{\# of iterations}) \times (\text{\# of bins}) \times \Pr(\mathcal{E}_b)$$

The number of bins is at most $\eta K$ for some constant $\eta$ and the number of iterations is at most $CK$ (at least one edge is peeled off at each iteration in the worst case). Hence, $\Pr(\mathcal{E}) \leq \eta C K^2 \Pr(\mathcal{E}_b)$. In order to satisfy $\Pr(\mathcal{E}) \leq O(1/K)$, we need to show that $\Pr(\mathcal{E}_b) \leq O(1/K^3)$.

We already showed in Lemma C.8 that we can achieve singleton identification under noise with vanishing error $O(1/K^3)$ with a delay matrix $\mathbf{D} \in \mathbb{Z}_2^{P_1 \times n}$.

To achieve type detection, we construct another pair of delay matrices $\mathbf{D}^1 \in \mathbb{Z}_2^{P_2 \times n}$ and $\mathbf{D}^2 \in \mathbb{Z}_2^{P_2 \times n}$. We will choose $\mathbf{D}^1$ and $\mathbf{D}^2$ such that each element is drawn i.i.d. as a $\mathrm{Bern}\left((1/2)^{1/t}\right)$. We denote

the $i^{th}$ row of $\mathbf{D}^1$ as $\mathbf{d}_i^1$ and denote the $i^{th}$ row of $\mathbf{D}^2$ as $\mathbf{d}_i^2$. Then, with these delay matrices, we can obtain observations of the form

$$U_i^1(\mathbf{k}) = \sum_{\substack{\mathbf{Hr}=\mathbf{k} \\ \mathbf{r} \leq \overline{\mathbf{d}}_i^1}} F(\mathbf{r}) + Z_i(\mathbf{k})$$

$$U_i^2(\mathbf{k}) = \sum_{\substack{\mathbf{Hr}=\mathbf{k} \\ \mathbf{r} \leq \overline{\mathbf{d}}_i^2}} F(\mathbf{r}) + Z_i(\mathbf{k}).$$

Note that we can represent these observations as

$$\mathbf{U}^1 = \mathbf{S}^1 \boldsymbol{\alpha} + \mathbf{W}^1$$
$$\mathbf{U}^2 = \mathbf{S}^2 \boldsymbol{\alpha} + \mathbf{W}^2$$

with $\mathbf{W}^1, \mathbf{W}^2 \sim \mathcal{N}(0, \sigma^2 \mathbf{I})$, a $\boldsymbol{\alpha}$ vector with entries $F(\mathbf{r})$ for coefficients in the set and binary signature matrices $\mathbf{S}^1, \mathbf{S}^2$ with entries indicating the subsets of coefficients included in each sum.

Then, we subtract these observations to obtain a single observation $\mathbf{U} = \mathbf{U}^1 - \mathbf{U}^2$ which can be written as

$$\mathbf{U} = \mathbf{S} \boldsymbol{\alpha} + \mathbf{W}$$

with $\mathbf{W} \sim \mathcal{N}(0, 2\sigma^2 \mathbf{I})$ and $\mathbf{S} = \mathbf{S}^1 - \mathbf{S}^2$. This construction allows us to show that the columns of $\mathbf{S}$ are sufficiently incoherent and hence we can correctly perform identification.

**Lemma C.9.** *For any fixed SNR, taking $\mathbf{D}_c^1$ and $\mathbf{D}_c^2$ such that each element is* $\mathrm{Bern}\left((1/2)^{1/t}\right)$*, and $t = \Theta(n^\alpha)$ for $\alpha \in (0, 1/2)$ and taking $P_2 = O(t \log(n))$ suffices to ensure that the probability $\Pr(\mathcal{E}_b)$ for an arbitrary bin can be upper bounded as $\Pr(\mathcal{E}_b) \leq O(1/K^3)$.*

*Proof.* In the following, we prove that $\Pr(\mathcal{E}_b) \leq O(1/K^3)$ holds using the observation model. We consider separate cases where the bin in consideration is fixed as a zeroton, singleton, or multiton.

The error probability $\Pr(\mathcal{E}_b)$ for an arbitrary bin can be upper bounded as

$$\Pr(\mathcal{E}_b) \leq \sum_{\mathcal{F} \in \{\mathcal{H}_Z, \mathcal{H}_M\}} \Pr(\mathcal{F} \leftarrow \mathcal{H}_S(\mathbf{r}, F(\mathbf{r})))$$
$$+ \sum_{\mathcal{F} \in \{\mathcal{H}_Z, \mathcal{H}_M\}} \Pr(\mathcal{H}_S(\widehat{\mathbf{r}}, \widehat{F}(\mathbf{r})) \leftarrow \mathcal{F})$$
$$+ \Pr(\mathcal{H}_S(\widehat{\mathbf{r}}, \widehat{F}(\mathbf{r})) \leftarrow \mathcal{H}_S(\mathbf{r}, F(\mathbf{r})))$$

above, each of these events should be read as:

1. $\{\mathcal{F} \leftarrow \mathcal{H}_S(\mathbf{r}, F(\mathbf{r}))\}$: missed verification in which the singleton verification fails when the ground truth is in fact a singleton.

2. $\{\mathcal{H}_S(\widehat{\mathbf{r}}, \widehat{F}(\mathbf{r})) \leftarrow \mathcal{F}\}$: false verification in which the singleton verification is passed when the ground truth is not a singleton.

3. $\{\mathcal{H}_S(\widehat{\mathbf{r}}, \widehat{F}(\mathbf{r})) \leftarrow \mathcal{H}_S(\mathbf{r}, F(\mathbf{r}))\}$: crossed verification in which a singleton with a wrong index-value pair passes the singleton verification when the ground truth is another singleton pair.

We can upper-bound each of these error terms using Propositions C.10, C.11, and C.12. Note that all upper-bound terms decay exponentially with $P_2$ except for the term $\Pr(\widehat{\mathbf{r}} \neq \mathbf{r}) \leq O(1/K^3)$.

We use Theorem F.3 to show that we can achieve $\Pr(\widehat{\mathbf{r}} \neq \mathbf{r}) \leq O(1/K^3)$ if we choose $P_1 = O(t \log n)$. Since all other error probabilities decay exponentially with $P_2$, it is clear that if $P_2$ is chosen as $P_2 = O(t \log n)$, the error probability can be bounded as $\Pr(\mathcal{E}_b) \leq O(1/K^3)$.

$\square$

**Proposition C.10** (False Verification Rate). *For $0 < \gamma < \frac{\eta}{4}\mathrm{SNR}$, the false verification rate for each bin hypothesis satisfies:*

$$\Pr(\mathcal{H}_S(\widehat{\mathbf{r}}, \widehat{F}(\widehat{\mathbf{r}})) \leftarrow \mathcal{H}_Z) \leq e^{-\frac{P_2}{2}(\sqrt{1+2\gamma}-1)^2},$$

$$\Pr(\mathcal{H}_S(\widehat{\mathbf{r}}, \widehat{F}(\widehat{\mathbf{r}})) \leftarrow \mathcal{H}_M) \leq e^{-\frac{P_2\gamma^2}{4(1+4\gamma)}} + K^2 e^{-\epsilon\left(1-\frac{4\gamma\nu^2}{\rho^2}\right)^2 P_2},$$

*where $P_2$ is the number of the random offsets.*

*Proof.* The probability of detecting a zeroton as a singleton can be upper-bounded by the probability of a zeroton failing the zeroton verification. This means

$$\Pr(\mathcal{H}_S(\widehat{\mathbf{r}}, \widehat{F}(\widehat{\mathbf{r}})) \leftarrow \mathcal{H}_Z) \leq \Pr\left(\frac{1}{P_2}\|\mathbf{W}\|^2 \geq (1+\gamma)\nu^2\right)$$

$$\leq e^{-\frac{P_2}{4}(\sqrt{1+2\gamma}-1)^2},$$

by noting that $\mathbf{W} \sim \mathcal{N}(0, \nu^2\mathbf{I})$ and applying Lemma C.13.

On the other hand, given some multiton observation $\mathbf{U} = \mathbf{S}\boldsymbol{\alpha} + \mathbf{W}$, the probability of detecting it as a singleton with index-value pair $(\widehat{\mathbf{r}}, \widehat{F}(\widehat{\mathbf{r}}))$ can be written as

$$\Pr(\mathcal{H}_S(\widehat{\mathbf{r}}, \widehat{F}(\widehat{\mathbf{r}})) \leftarrow \mathcal{H}_M) = \Pr\left(\frac{1}{P_2}\left\|\mathbf{U} - \widehat{F}(\widehat{\mathbf{r}})\mathbf{s}_{\widehat{\mathbf{r}}}\right\|^2 \leq (1+\gamma)\nu^2\right) =$$

$$\Pr\left(\frac{1}{P_2}\left\|\mathbf{g} + \mathbf{v}\right\|^2 \leq (1+\gamma)\nu^2\right),$$

where $\mathbf{g} := \mathbf{S}(\boldsymbol{\alpha} - \widehat{F}(\widehat{\mathbf{r}})\mathbf{e}_{\widehat{\mathbf{r}}})$ and $\mathbf{v} := \mathbf{W}$. Then, we can upper bound this probability as

$$\Pr\left(\frac{1}{P_2}\left\|\mathbf{g} + \mathbf{v}\right\|^2 \leq (1+\gamma)\nu^2 \,\middle|\, \frac{\|\mathbf{g}\|^2}{P_2} \geq 2\gamma\nu^2\right) + \Pr\left(\frac{\|\mathbf{g}\|^2}{P_2} \leq 2\gamma\nu^2\right).$$

To upper bound the first term, we use Lemma C.13. Note that the first term is conditioned on the event $\|\mathbf{g}\|^2/P_2 \geq 2\gamma\nu^2$, thus the normalized non-centrality parameter satisfies $\theta_0 \geq 2\gamma$. As a result, we can use Lemma C.13 by letting $\tau_2 = (1+\gamma)\nu^2$. Then, the first term is upper bounded by $\exp\{-(P_2\gamma^2)/(4(1+4\gamma))\}$. To analyze the second term, we let $\boldsymbol{\beta} = \boldsymbol{\alpha} - \widehat{F}(\widehat{\mathbf{r}})\mathbf{e}_{\widehat{\mathbf{r}}}$ and write $\mathbf{g} = \mathbf{S}\boldsymbol{\beta}$. Denoting its support as $\mathcal{L} := \mathrm{supp}(\boldsymbol{\beta})$, we can further write $\mathbf{S}\boldsymbol{\beta} = \mathbf{S}_{\mathcal{L}}\boldsymbol{\beta}_{\mathcal{L}}$ where $\mathbf{S}_{\mathcal{L}}$ is the sub-matrix of $\mathbf{S}$ consisting of the columns in $\mathcal{L}$ and $\boldsymbol{\beta}_{\mathcal{L}}$ is the sub-vector consisting of the elements in $\mathcal{L}$. Then, we consider two scenarios:

- The multiton size is a constant, i.e., $|\mathcal{L}| = L = O(1)$. In this case, we have

$$\lambda_{\min}(\mathbf{S}_{\mathcal{L}}^\top\mathbf{S}_{\mathcal{L}})\|\boldsymbol{\beta}_{\mathcal{L}}\|^2 \leq \|\mathbf{S}_{\mathcal{L}}\boldsymbol{\beta}_{\mathcal{L}}\|^2$$

  Using $\|\boldsymbol{\beta}_{\mathcal{L}}\|^2 \geq L\rho^2$, the probability can be bounded as

$$\Pr\left(\frac{\|\mathbf{g}\|^2}{P_2} \leq 2\gamma\nu^2\right) \leq \Pr\left(\lambda_{\min}\left(\frac{1}{P_2}\mathbf{S}_{\mathcal{L}}^\top\mathbf{S}_{\mathcal{L}}\right) \leq \frac{2\gamma\nu^2}{L\rho^2}\right)$$

  On the other hand, using Lemma C.14 with the selection $\beta = 1/2$ and $\eta = \frac{1}{1+2L}\left(\frac{1}{2} - \frac{2\gamma\nu^2}{L\rho^2}\right)$, we have

$$\Pr\left(\frac{\|\mathbf{g}\|^2}{P_2} \leq 2\gamma\nu^2\right) \leq 2L^2 e^{-\frac{P_2}{2(1+2L)^2}\left(\frac{1}{2} - \frac{2\gamma\nu^2}{L\rho^2}\right)^2}.$$

  which holds as long as $\gamma < L\rho^2/(4\nu^2) = \frac{L\eta}{4}\mathrm{SNR}$.

- The multiton size grows asymptotically with respect to $K$, i.e., $|\mathcal{L}| = L = \omega(1)$. As a result, the vector of random variables $\mathbf{g} = \mathbf{S}_{\mathcal{L}}\boldsymbol{\beta}_{\mathcal{L}}$ becomes asymptotically Gaussian due to the central limit theorem with zero mean and a covariance

$$\mathbb{E}[\mathbf{g}\mathbf{g}^{\mathrm{H}}] = \frac{1}{2}L\rho^2\mathbf{I}$$

Therefore, by Lemma C.13, we have

$$\Pr\left(\frac{\|\mathbf{g}\|^2}{P_2} \leq 2\gamma\nu^2\right) \leq e^{-\frac{P_2}{2}\left(1-\frac{\gamma\nu^2}{L\rho^2}\right)}$$

which holds as long as $\gamma < L\rho^2/\nu^2 = L\eta\mathrm{SNR}$.

By combining the results from both cases, there exists some absolute constant $\epsilon > 0$ such that

$$\Pr\left(\frac{\|\mathbf{g}\|^2}{P_2} \leq 2\gamma\nu^2\right) \leq K^2 e^{-\epsilon\left(1-\frac{4\gamma\nu^2}{\rho^2}\right)^2 P_2}$$

as long as $\gamma < \rho^2/(4\nu^2) = \frac{\eta}{4}\mathrm{SNR}$. $\qquad\qquad\square$

**Proposition C.11** (Missed Verification Rate). *For $0 < \gamma < \frac{\eta}{2}\mathrm{SNR}$, the missed verification rate for each bin hypothesis satisfies*

$$\Pr(\mathcal{H}_Z \leftarrow \mathcal{H}_S(\mathbf{r}, F[\mathbf{r}])) \leq e^{-\frac{P_2}{4}\frac{(\rho^2/\nu^2-\gamma)^2}{1+2\rho^2/\nu^2}}$$

$$\Pr(\mathcal{H}_M \leftarrow \mathcal{H}_S(\mathbf{r}, F[\mathbf{r}])) \leq e^{-\frac{P_2}{4}(\sqrt{1+2\gamma}-1)^2} + 2e^{-\frac{\rho^2}{2\nu^2}P_2} + 2\Pr(\widehat{\mathbf{r}} \neq \mathbf{r})$$

*where $P_2$ is the number of the random offsets.*

*Proof.* The probability of detecting a singleton as a zeroton can be upper bounded by the probability of a singleton passing the zeroton verification. Hence, by noting that $\mathbf{W} \sim \mathcal{N}(0, \nu^2\mathbf{I})$ and applying Lemma C.13,

$$\Pr(\mathcal{H}_Z \leftarrow \mathcal{H}_S(\mathbf{r}, F[\mathbf{r}]))$$
$$\leq \Pr\left(\frac{1}{P_2}\|F[\mathbf{r}]\mathbf{s}_{\mathbf{r}} + \mathbf{W}\|^2 \leq (1+\gamma)\nu^2\right)$$
$$\leq e^{-\frac{P_2}{4}\frac{(\rho^2/\nu^2-\gamma)^2}{1+2\rho^2/\nu^2}}.$$

which holds as long as $\gamma < \rho^2/\nu^2 = \eta\mathrm{SNR}$.

On the other hand, the probability of detecting a singleton as a multiton can be written as the probability of failing the singleton verification step for some index-value pair $(\widehat{\mathbf{r}}, \widehat{F}[\widehat{\mathbf{r}}])$. Hence, we can write

$$\Pr(\mathcal{H}_M \leftarrow \mathcal{H}_S(\mathbf{r}, F[\mathbf{r}])) = \Pr\left(\frac{1}{P_2}\left\|\mathbf{U} - \widehat{F}[\widehat{\mathbf{r}}]\mathbf{s}_{\widehat{k}}\right\|^2 \geq (1+\gamma)\nu^2\right)$$
$$\leq \Pr\left(\frac{1}{P_2}\left\|\mathbf{U} - \widehat{F}[\widehat{\mathbf{r}}]\mathbf{s}_{\widehat{k}}\right\|^2 \geq (1+\gamma)\nu^2\Big|\widehat{F}[\widehat{\mathbf{r}}] = F[\mathbf{r}] \wedge \widehat{\mathbf{r}} = \mathbf{r}\right)$$
$$+ \Pr(\widehat{F}[\widehat{\mathbf{r}}] \neq F[\mathbf{r}] \vee \widehat{\mathbf{r}} \neq \mathbf{r}).$$

Then, using Lemma C.13, the first term is upper-bounded as

$$\Pr\left(\frac{1}{P_2}\left\|\mathbf{U} - \widehat{F}[\widehat{\mathbf{r}}]\mathbf{s}_{\widehat{k}}\right\|^2 \geq (1+\gamma)\nu^2\Big|\widehat{F}[\widehat{\mathbf{r}}] = F[\mathbf{r}] \wedge \widehat{\mathbf{r}} = \mathbf{r}\right) \leq \Pr\left(\frac{1}{P_2}\|\mathbf{W}\|^2 \geq (1+\gamma)\nu^2\right)$$
$$\leq e^{-\frac{P_2}{4}(\sqrt{1+2\gamma}-1)^2}.$$

On the other hand, the second term can be bounded as

$$\Pr(\widehat{F}[\widehat{\mathbf{r}}] \neq F[\mathbf{r}] \vee \widehat{\mathbf{r}} \neq \mathbf{r}) \leq \Pr(\widehat{F}[\widehat{\mathbf{r}}] \neq F[\mathbf{r}]) + \Pr(\widehat{\mathbf{r}} \neq \mathbf{r})$$
$$= \Pr(\widehat{F}[\widehat{\mathbf{r}}] \neq F[\mathbf{r}]|\widehat{\mathbf{r}} \neq \mathbf{r})\Pr(\widehat{\mathbf{r}} \neq \mathbf{r})$$
$$+ \Pr(\widehat{F}[\widehat{\mathbf{r}}] \neq F[\mathbf{r}]|\widehat{\mathbf{r}} = \mathbf{r})\Pr(\widehat{\mathbf{r}} = \mathbf{r})$$
$$+ \Pr(\widehat{\mathbf{r}} \neq \mathbf{r})$$
$$\leq \Pr(\widehat{F}[\widehat{\mathbf{r}}] \neq F[\mathbf{r}]|\widehat{\mathbf{r}} = \mathbf{r}) + 2\Pr(\widehat{\mathbf{r}} \neq \mathbf{r})$$

The first term is the error probability of a BPSK signal with amplitude $\rho$, and it can be bounded as

$$\Pr(\widehat{F}[\widehat{\mathbf{r}}] \neq F[\mathbf{r}]|\widehat{\mathbf{r}} = \mathbf{r}) \leq 2e^{-\frac{\rho^2}{2\nu^2}P_2}$$

$\square$

**Proposition C.12** (Crossed Verification Rate). *For $0 < \gamma < \frac{\eta}{2}\text{SNR}$, the crossed verification rate for each bin hypothesis satisfies*

$$\Pr(\mathcal{H}_S(\widehat{\mathbf{r}}, \widehat{F}[\widehat{\mathbf{r}}]) \leftarrow \mathcal{H}_S(\mathbf{r}, F[\mathbf{r}])) \leq e^{-\frac{P_2\gamma^2}{4(1+4\gamma)}} + Ke^{-\epsilon\left(1-\frac{4\gamma\nu^2}{\rho^2}\right)^2 P_2} + K^2 e^{-\epsilon\left(1-\frac{4\gamma\nu^2}{\rho^2}\right)^2 P_2^2/t}.$$

*where $P_2$ is the number of the random offsets.*

*Proof.* This error event can only occur if a singleton with index-value pair $(\mathbf{r}, F[\mathbf{r}])$ passes the singleton verification step for some index-value pair $(\widehat{\mathbf{r}}, \widehat{F}[\widehat{\mathbf{r}}])$ such that $\mathbf{r} \neq \widehat{\mathbf{r}}$. Hence,

$$\Pr(\mathcal{H}_S(\widehat{\mathbf{r}}, \widehat{F}[\widehat{\mathbf{r}}]) \leftarrow \mathcal{H}_S(\mathbf{r}, F[\mathbf{r}]))$$

$$\leq \Pr\left(\frac{1}{P_2}\|F[\mathbf{r}]\mathbf{s_r} - \widehat{F}[\widehat{\mathbf{r}}]\mathbf{s_{\widehat{r}}} + \mathbf{W}\|^2 \leq (1+\gamma)\nu^2\right)$$

$$= \Pr\left(\frac{1}{P_2}\|\mathbf{S}\boldsymbol{\beta} + \mathbf{W}\|^2 \leq (1+\gamma)\nu^2\right)$$

$$= \Pr\left(\frac{1}{P_2}\|\mathbf{S}\boldsymbol{\beta} + \mathbf{W}\|^2 \leq (1+\gamma)\nu^2 \middle| \|\mathbf{S}\boldsymbol{\beta}\|^2 \geq 2\gamma\nu^2\right)$$

$$+ \Pr\left(\|\mathbf{S}\boldsymbol{\beta}\|^2 \leq 2\gamma\nu^2\right)$$

where $\boldsymbol{\beta}$ is a 2-sparse vector with non-zero entries from $\{\rho, -\rho\}$. Using Lemma C.13, the first term is upper-bounded as

$$\Pr\left(\frac{1}{P_2}\|\mathbf{S}\boldsymbol{\beta} + \mathbf{W}\|^2 \leq (1+\gamma)\nu^2 \middle| \|\mathbf{S}\boldsymbol{\beta}\|^2 \geq 2\gamma\nu^2\right) \leq e^{-\frac{P_2\gamma^2}{4(1+4\gamma)}}.$$

By Lemma C.14, the second term is upper bounded as

$$\Pr\left(\|\mathbf{S}\boldsymbol{\beta}\|^2 \leq 2\gamma\nu^2\right) \leq 8e^{-\frac{P_2}{50}\left(\frac{1}{2} - \frac{\gamma\nu^2}{L\rho^2}\right)^2}$$

which holds as long as $\gamma < \rho^2/(2\nu^2) = \frac{\eta}{2}\text{SNR}$.

$\square$

**Lemma C.13** (Non-central Tail Bounds (Lemma 11 in [13])). *Given any $\mathbf{g} \in \mathbb{R}^P$ and a Gaussian vector $\mathbf{v} \sim \mathcal{N}(0, \nu^2\mathbf{I})$, the following tail bounds hold:*

$$\Pr\left(\frac{1}{P}\|\mathbf{g} + \mathbf{v}\|^2 \geq \tau_1\right) \leq e^{-\frac{P}{4}(\sqrt{2\tau_1/\nu^2 - 1} - \sqrt{1+2\theta_0})^2}$$

$$\Pr\left(\frac{1}{P}\|\mathbf{g} + \mathbf{v}\|^2 \leq \tau_2\right) \leq e^{-\frac{P}{4}\frac{\left(1+\theta_0 - \tau_2/\nu^2\right)^2}{1+2\theta_0}}$$

*for any $\tau_1$ and $\tau_2$ that satisfy $\tau_1 \geq \nu^2(1+\theta_0) \geq \tau_2$ where*

$$\theta_0 := \frac{\|\mathbf{g}\|^2}{P\nu^2}$$

*is the normalized non-centrality parameter.*

**Lemma C.14.** *Suppose $\beta = \Theta(1)$, $\eta = \Omega(1)$, and $t = \Theta(n^\alpha)$ for some $\alpha \in (0, 1/2)$. Then, there exists some $n_0$ such that for all $n \geq n_0$, we have*

$$\Pr\left(\lambda_{\min}\left(\frac{1}{P_2}\mathbf{S}_{\mathcal{L}}^\top \mathbf{S}_{\mathcal{L}}\right) \leq 2\beta(1-\beta) - (2L+1)\eta\right) \leq 2L^2 \exp\left(-\frac{\eta^2}{2}P_2\right).$$

*Proof.* For any $\mathbf{r}$ sampled uniformly from vectors up to degree $t$, the probability that it will have degree $0 \leq k \leq t$ can be written as

$$\Pr\left(|\mathbf{r}| = k\right) = \frac{\binom{n}{k}}{\sum_{k=1}^{t} \binom{n}{k}}$$

We know that the entries of $\mathbf{s_r}$ are given as $(\mathbf{s_r^1})_i = \mathbb{1}\left\{\mathbf{r} \leq \bar{\mathbf{d}}_i^1\right\}$ and $(\mathbf{s_r^2})_i = \mathbb{1}\left\{\mathbf{r} \leq \bar{\mathbf{d}}_i^2\right\}$. Therefore,

$$\begin{aligned}
\Pr\left((\mathbf{s_r^1})_i = 1\right) &= \Pr\left(d_{ij}^1 = 0, \forall j \in \mathrm{supp}(\mathbf{r})\right) \\
&= \sum_{k=1}^{t} \Pr\left(d_{ij}^1 = 0, \forall j \in \mathrm{supp}(\mathbf{r}) | |\mathbf{r}| = k\right) \Pr\left(|\mathbf{r}| = k\right) \\
&= \frac{\sum_{k=1}^{t} \binom{n}{k} \beta^{k/t}}{\sum_{k=1}^{t} \binom{n}{k}}. \\
&=: g(t, n)
\end{aligned}$$

With $\beta = \Theta(1)$ and $t = \Theta(n^\alpha)$ for $\alpha \in (0, 1/2)$, we can show that $\lim_{n\to\infty} g(t, n) = \beta$. Therefore, there exists some $n_0$ such that $|\Pr\left((\mathbf{s_r^1})_i = 1\right) - \beta| \leq \eta$ for all $n \geq n_0$. For the rest of the proof, let $g = \Pr\left((\mathbf{s_r^1})_i = 1\right)$ and assume $|g - \beta| \leq \eta$.

Then, recalling $(\mathbf{s_r})_i = (\mathbf{s_r^1})_i - (\mathbf{s_r^2})_i$, the distribution for each entry of $\mathbf{s_r}$ can be written as

$$\Pr\left((\mathbf{s_r})_i = 1\right) = \Pr\left((\mathbf{s_r})_i = -1\right) = g(1 - g).$$

Hence, using Hoeffding's inequality, we obtain

$$\Pr\left(\frac{1}{P_2}\mathbf{s_r}^\top \mathbf{s_r} \leq 2\beta(1 - \beta) - \eta\right) \leq \Pr\left(\frac{1}{P_2}\mathbf{s_r}^\top \mathbf{s_r} \leq 2g(1 - g) - \eta\right) \leq \exp\left(-\frac{\eta^2}{2}P_2\right).$$

Furthermore, the conditional probability of another vector $\mathbf{m} \neq \mathbf{r}$ being included in test $i$ is given by

$$\begin{aligned}
\Pr\left((\mathbf{s_m^1})_i = 1 | (\mathbf{s_r^1})_i = 1, |\mathbf{r}| = k\right) &= \Pr\left(d_{ij} = 0, \forall j \in \mathrm{supp}(\mathbf{m}) \setminus \mathrm{supp}(\mathbf{r}) | |\mathbf{r}| = k\right) \\
&= \sum_{\ell=0}^{t} \left(\beta^{1/t}\right)^\ell \left(1 - \frac{k}{n}\right)^\ell \left(\frac{k}{n}\right)^{t-\ell} \\
&= \left(\frac{k}{n} + \left(1 - \frac{k}{n}\right)\beta^{1/t}\right)^t \\
&=: f(t, n, k).
\end{aligned}$$

With $\beta = \Theta(1)$ and $t = \Theta(n^\alpha)$ for $\alpha \in (0, 1)$, for any $k \leq t$, we can show that $\lim_{n\to\infty} f(t, n, k) = \beta$. Therefore, there exists some $n_0$ such that $|\Pr\left((\mathbf{s_m^1})_i = 1 | (\mathbf{s_r^1})_i = 1\right) - \beta| \leq \eta$ for all $n \geq n_0$. For the rest of the proof, let $f = \Pr\left((\mathbf{s_m^1})_i = 1 | (\mathbf{s_r^1})_i = 1\right)$ and assume $|f - \beta| \leq \eta$.

On the other hand,

$$\begin{aligned}
\Pr\left((\mathbf{s_m})_i(\mathbf{s_r})_i = 1\right) &= 2fg\left[1 - g - (1 - f)g\right] \\
\Pr\left((\mathbf{s_m})_i(\mathbf{s_r})_i = -1\right) &= 2\left[(1 - f)g\right]^2
\end{aligned}$$

As a result, we have

$$\mathbb{E}[(\mathbf{s_m})_i(\mathbf{s_r})_i] = 2g(f - g).$$

Since $\lim_{n\to\infty} \mathbb{E}[(\mathbf{s_m})_i(\mathbf{s_r})_i] = 0$, there exists some $n_0$ such that $-\eta \leq \mathbb{E}[(\mathbf{s_m})_i(\mathbf{s_r})_i] \leq \eta$ for all $n \geq n_0$. For the rest of the proof assume $-\eta \leq \mathbb{E}[(\mathbf{s_m})_i(\mathbf{s_r})_i] \leq \eta$. As a result, we can write

$$\Pr\left(\frac{1}{P_2}|\mathbf{s_r}^\top \mathbf{s_m}| \geq 2\eta\right) \leq \Pr\left(|\mathbf{s_r}^\top \mathbf{s_m} - P_2\mathbb{E}[(\mathbf{s_m})_i(\mathbf{s_r})_i]| \geq P_2\eta\right) \leq \exp\left(-\frac{\eta^2}{2}P_2\right).$$

By Gershgorin Circle Theorem, the minimum eigenvalue of $\frac{1}{P_2}\mathbf{S}_{\mathcal{L}}^{\top}\mathbf{S}_{\mathcal{L}}$ is lower bounded as

$$\lambda_{\min}\left(\frac{1}{P_2}\mathbf{S}_{\mathcal{L}}^{\top}\mathbf{S}_{\mathcal{L}}\right) \geq \frac{1}{P_2}\min_{\mathbf{r}\in\mathcal{L}}\left(|\mathbf{s}_{\mathbf{r}}^{\top}\mathbf{s}_{\mathbf{r}}| - \sum_{\substack{\mathbf{m}\in\mathcal{L}\\ \mathbf{m}\neq\mathbf{r}}}|\mathbf{s}_{\mathbf{r}}^{\top}\mathbf{s}_{\mathbf{m}}|\right).$$

Lastly, we apply a union bound over all $(\mathbf{r},\mathbf{m})$ pairs to obtain

$$\Pr\left(\lambda_{\min}\left(\frac{1}{P_2}\mathbf{S}_{\mathcal{L}}^{\top}\mathbf{S}_{\mathcal{L}}\right) \leq 2\beta(1-\beta) - (2L+1)\eta\right) \leq 2L^2\exp\left(-\frac{\eta^2}{2}P_2\right).$$

$\square$

## D    Worst-Case Time Complexity

In this section, we discuss the computational complexity of Algorithm 1, which is broken down into the following parts:

**Computing Samples**    Computing samples for one sapling matrix requires computing the row-span of $\mathbf{H}_c$, which can be computed in $n2^b$ operations. Then for each sample, we must take the bit-wise and with each row of the delay matrix, so the total complexity is: $Cn2^bP$.

**Taking Small Mobius Transform**    Computing the Mobius transform for each of the $CP$ subsampled functions is $CPb2^b$.

**Singleton Detection**    To detect each singleton requires computing $\mathbf{y}$. This requires $P$ divisions for each of the $C2^b$ bins, for a total of $CP2^b$ operations.

**Singleton Identification**    To identify each singleton requires different complexity for our different assumptions.

1. In the case of uniformly distributed interactions, singleton detection is $O(1)$, since $\mathbf{y} = \mathbf{k}^*$ immediately, so doing this for each singleton makes the total complexity $CK$.
2. In the noiseless low degree case decoding $\mathbf{k}^*$ from $\mathbf{y}$ is $\text{poly}(n)$, so for each singleton the complexity is $CK\,\text{poly}(n)$

**Message Passing**    In the worst case, we peel exactly one singleton per iteration, resulting in $CK$ subtractions (the above singleton identification bounds already take into account the need to re-do singleton identification).

Thus in the case of uniformly distributed and low degree interactions respectively, the complexity is:

$$\begin{aligned}
\text{Uniform distributed noiseless time complexity} \quad &= \quad O(CPn2^b + CPb2^b + CK)\\
&= \quad O(CPnK)\\
&= \quad O(n^2K).
\end{aligned}$$

$$\begin{aligned}
\text{Low degree (noisy) time complexity} \quad &= \quad O(CPn2^b + CPb2^b + CK\,\text{poly}(n))\\
&= \quad O(CP\,\text{poly}(n)K)\\
&= \quad O(\text{poly}(n)K).
\end{aligned}$$

## E    Additional Simulations

In this section, we present some additional simulations that did not fit in the body of the manuscript. Fig 15 and 16. Plot the runtime of SMT vs. $n$ under both of our assumptions. In both cases we observe excellent scaling with $n$. We note that our low degree setting has a higher fixed cost since we

are using linear programming to solve our group testing problem and the solver appears to have some non-trivial fixed time cost.

Fig. 17 plots the perfect reconstruction percentage against $n$ and sample complexity. We also observe a phase transition, however, the phase threshold appears very insensitive to $n$, as expected, since our sample complexity requirement is growing like $\log(n)$, and we are already plotting on a log scale.

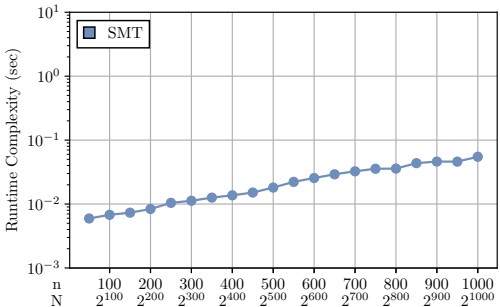

Figure 15: Time complexity of SMT under Assumption 2.1. The parameter $K$ is fixed and we plot the runtime v.s. $n$. our algorithm remains possible to run for $n = 1000$ where other competitors fail.

Figure 16: Time complexity of SMT under assumption 2.2. The parameters $K$ and $t$ are fixed and we plot the runtime v.s. $n$. Our theory says we have a $\mathrm{poly}(n)$ complexity. In practice, for reasonable $n$ our algorithm is running quickly.

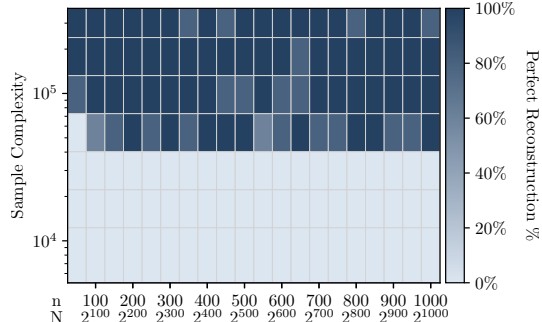

Figure 17: Perfect reconstruction percentage plotted against sample complexity and $n$ under Assumption 2.2. Holding $C = 3$, we scale $b$ to increase the sample complexity. We observe that the number of samples required to achieve perfect reconstruction is scaling linearly is very insensitive to $n$ as predicted. We also include $N = 2^n$ on the bottom axis, which is the total number of interactions. In this regime we do not appear to consistently maintain zero error. This could be due to the fact that the asymptotic behaviour of group testing might not yet be fuly realized in the regime with $n \leq 1000$.

## F    Group Testing

### F.1    Group Testing Achievability Results From Literature

**Theorem F.1** (Part of Theorem 4.1 and 4.2 in [15]). *Asymptotic Rate 1 Noiseless Group Testing: Consider a noiseless group testing problem with $t = \Theta(n^\theta)$ defects out of $n$ elements. We define the rate of a group testing procedure as:*

$$R := \frac{\log \binom{n}{t}}{T} \tag{74}$$

*where $T$ is the number of tests performed by the group testing procedure. For an i.i.d. Bernoulli design matrix, for $\theta \in [0, 1/3]$, in the limit as $n \to \infty$, a rate $R^*_{BERN} = 1$ is achievable with vanishing error. Furthermore, for the constant column-weight design matrix, for $\theta \in [0, 0.409]$ a rate $R^*_{CCW} = 1$ is achievable with vanishing error.*

**Theorem F.2** ([42, 54]). *Noiseless Group Testing: Consider the noiseless non-adaptive group testing setup with $t = |\mathbf{k}|$ defects out of $n$ items, with $t$ scaling arbitrarily in $n$. Let $\hat{\mathbf{k}}$ be the output*

*of a group testing decoder and let $T^* = \Theta\left(\min\left\{t\log(n), n\right\}\right)$. Then there exists a strategy using $T \le (1+\epsilon)T^*$ such that in the limit as $n \to \infty$ we have:*

$$\Pr\left(\hat{\mathbf{k}} \ne \mathbf{k}\right) \to 0. \tag{75}$$

*Furthermore, there is a $\mathrm{poly}(n)$ algorithm for computing $\hat{\mathbf{k}}$. From [54], for $t = o(n)$ we can achieve:*

$$\Pr\left(\hat{\mathbf{k}} \ne \mathbf{k}\right) \le n^{-\delta}. \tag{76}$$

*with number of tests $T = O((1+\delta)t\log(n))$.*

Note that the above error rate is not a state-of-art result, but suffices in this case for our proof, and is very convenient in its form.

**Theorem F.3** ([43]). *Noisy Group Testing Under General Binary Noise: Consider the general binary noisy group testing setup with crossover probabilities $p_{10}$ and $p_{01}$. We use i.i.d Bernoulli testing with parameter $\nu > 0$. There are a total of $|\mathbf{k}| = t = \Theta(n^\theta)$ defects, where $\theta \in (0,1)$. Let $T^* = \max\left\{T_1^{(D)}, T_1^{(ND)}, T_2^{(D)}, T_2^{(ND)}\right\}$, where we have*

$$T_1^{(D)} = \frac{1}{\nu p_{10} D_{(\alpha/p_{10})}} t\log(t), \tag{77}$$

$$T_1^{(ND)} = \frac{1}{\nu w D(\alpha/w)} t\log(n), \tag{78}$$

$$T_2^{(D)} = \frac{1}{\nu e^{-\nu}(1 - p_{10}) D(\beta/p_{10})} t\log(t), \tag{79}$$

$$T_2^{(ND)} = \frac{1}{\nu p_{01} D(\beta/p_{01})} t\log(n). \tag{80}$$

*where $D(x) = x\log(x) - x + 1$, and $w = (1 - p_{01})e^{-\nu} + p_{10}(1 - e^{-\nu})$. For any $\alpha \in (p_{10}, 1 - p_{01})$, $\beta \in (p_{01}, 1 - p_{10})$, there exist some number of tests $T < (1+\epsilon)T^*$ where the Noisy DD algorithm produces $\hat{\mathbf{k}}$ such that in the limit as $n \to \infty$ we have:*

$$\Pr\left(\hat{\mathbf{k}} \ne \mathbf{k}\right) \to 0. \tag{81}$$

The above result is state-of-art for noisy group testing and could be of interest generally for proving the type of results we have here, however, but for simplicity, we state a similar more compact result that suffices for our proofs in this paper.

**Theorem F.4** ([54]). *Let $|\mathbf{k}| = t = o(n)$, and consider an i.i.d. Bernoulli design group testing matrix. Further consider the binary symmetric noise model with crossover probability $q$. If we construct $\hat{\mathbf{k}}$ via the noisy column matching algorithm, we achieve:*

$$\Pr\left(\hat{\mathbf{k}} \ne \mathbf{k}\right) \le n^{-\beta}, \tag{82}$$

*with number of tests*

$$T = \frac{16(1 + \sqrt{\gamma})^2(1 + \beta)\ln(2)}{1 - e^{-2}(1 - 2q)^2} t\log(n). \tag{83}$$

*where $\gamma$ is a constant that depends on $\beta$.*

### F.2   Group Testing Implementation

We implement group testing via linear programming. As noted in [15], linear programming generally outperforms most other group testing algorithms in both the noisy and noiseless case. We use the

following linear program, to implement group testing.

$$\min_{\mathbf{k}, \boldsymbol{\xi}} \quad \sum_{i=1}^{n} k_i + \lambda \sum_{p=1}^{P} \xi_j$$

$$\text{s.t.} \quad k_i \geq 0$$
$$\xi_p \geq 0 \tag{84}$$
$$\xi_p \leq 1 \quad p \text{ s.t. } y_p = 1$$
$$\mathbf{d}_p^{\mathrm{T}} \mathbf{k} = \xi_p \quad p \text{ s.t. } y_p = 0$$
$$\mathbf{d}_p^{\mathrm{T}} \mathbf{k} + \xi_p \geq 1 \quad p \text{ s.t. } y_p = 1$$

## G   Impact Statement

Rigorous tools for understanding models can potentially profoundly increase trust in deep learning systems. If we can understand and reason for ourselves why a model is making a decision, we can put greater trust into those decisions. Furthermore, if we understand why a model is doing something that we believe is incorrect, we can better steer it towards doing what we believe is correct. This "steering" of model behavior is sometimes described as *alignment*, and is a critical task for addressing things like incorrect or misleading information generated by a model, or for address any undesirable biases. In terms of concerns, it is important to not misinterpret or over-interpret the interaction indices that come out of SMT. It could be the case that looking over some selection of interactions doesn't reveal the full picture, and leads one down an incorrect line of reasoning.

