# OpenReview forum: "Learning to Understand: Identifying Interactions via the Möbius Transform"
_NeurIPS.cc/2024/Conference — NeurIPS 2024 poster_

### Official Review · Reviewer_vNmZ · 2024-07-07

**Soundness:** 2
**Presentation:** 1
**Contribution:** 3
**Rating:** 5
**Confidence:** 5

**Summary:**

The paper present a way to compute high-order interactions using Mobius transformation.

**Strengths:**

The paper shows some interesting results on the sampling requirement for recovering the interactions among features, specially tailored for explainability with boolean functions (i.e., explanation game)

**Weaknesses:**

1. The paper is difficult to follow and does not have a clear storyline; most importantly, it is not clear how the significant interactions could be recovered at the end after obtaining the required samples

2. Aside from the clarification, though having an innovative approach and some interesting results, the method seems to have little applicability in real-world applications; one reason for doing so is one of the main assumptions of the method of having only K nonzero interactions (= K nonzero Mobius coefficient). For explainability, as defined, for instance, in SHAP, it is typically the case that we have many nonzero interactions since it is rare that the interactions among a feature subset will go to zero (given the way we compute the interactions). This implies that in most cases, we have a very large K of nonzero interactions, and this would make the sparse Mobius transformation impractical.

3. That being said, it is not clear how exactly the interactions are obtained; as the authors stated in the paper, the naive computation of all Mobius coefficients includes exponentially many parameters; given the samples, what interactions should be included as significant?

4. Last but not least, in explainability, we usually seek the most influential features and the most important interactions (rather than all feature importance or significant interactions); the paper seems to merely focus on computing all the significant interactions rather than the most important ones, and in explainability, this is not essentially sought and seems impractical due to many nonzero interactions.

Minor:
Some nations need more explanation as they are borrowed from other disciplines (as the authors also stated in the Introduction, the paper is multidisciplinary); For instance, the functional subsampling and aliasing that are the backbone of the sampling strategies, required more explanation to average reader of the paper, IMHO.

**Questions:**

how can the results of the paper be generalized to non-boolean functions? this would generalize the results from detecting interactions for explainability to a general case of interaction detection for any arbitrary dataset.

**Limitations:**

The assumption of having only K non-zero interactions, while in practice K is typically large for explainability.

---

> ### Author Rebuttal · Authors · 2024-08-05
>
> Thank you for your review. We believe that some aspects of this review may be based on misunderstandings, which may have resulted in some false or misleading statements in your review. In this rebuttal we will provide additional information and clarification that will hopefully resolve your concerns, and help underscore the significance of this work.
>
> **Point 1 and 3**: In your first point you said: "it is not clear how the significant interactions could be recovered at the end after obtaining the required samples". You also asked in your third point "That being said, it is not clear how exactly the interactions are obtained; as the authors stated in the paper, the naive computation of all Mobius coefficients includes exponentially many parameters; given the samples, what interactions should be included as significant?"
>
> A: The purpose of this manuscript is precisely to show how a small number of the interactions (which correspond to Mobius coefficients) can be recovered. A Mobius coefficient corresponds to an interaction, because it represents the joint effect of a given set of inputs that only occurs when all those given inputs are active.
>
> Theorem 1 shows that our algorithm recover the $K$ non-zero Mobius coefficients using the SMT algorithm. In this noise-free context "significant" means non-zero.
> Theorem $2$ considers the setting where we have $K$ large non-zero Mobius coefficients, but there is also noise, which corresponds to many relatively small interactions between inputs. In this context the SMT algorithm recovers the $K$ large interactions, which are the ones we call "significant".
>
> **Point 2**: You have also made the following claim. "Aside from the clarification, though having an innovative approach and some interesting results, the method seems to have little applicability in real-world applications; one reason for doing so is one of the main assumptions of the method of having only K nonzero interactions (= K nonzero Mobius coefficient). For explainability, as defined, for instance, in SHAP, it is typically the case that we have many nonzero interactions since it is rare that the interactions among a feature subset will go to zero (given the way we compute the interactions). This implies that in most cases, we have a very large K of nonzero interactions, and this would make the sparse Mobius transformation impractical."
>
> A: This is an important distinction. You are correct in your statement that in nearly all practical settings, there will be a large number of *nonzero* interactions (or Mobius coefficients).  However, what has been discovered by ourselves and many others is that a relatively small number of Mobius coefficients are needed to faithfully represent the functions learned by well-trained machine learning algorithms. For reference on a diverse set of machine learning algorithms and applications, we refer the review to Figures 2 and 6 in our paper, Figure 3 in [1], and Figure 3 in [2]. Furthermore, as shown Figure 2 in our paper and Figures 4, 7-9 in [1], these significant interactions (high magnitude Mobius coefficients) tend to be low-order. Under the interpretation that insignificant interactions are treated as noise (an interpretation also made by [1] and [2]), we provided in this work a noise-robust algorithm for extracting the K significant interactions up to order t, with the precise statement of this result in Theorem 5.2.
>
> [1] Ren et al. Where We Have Arrived in Proving the Emergence of Sparse Interaction Primitives in DNNs. ICLR, 2024.
>
> [2] Li et al. Does a Neural Network Really Encode Symbolic Concepts? ICML, 2023.
>
> **Point 4**: "Last but not least, in explainability, we usually seek the most influential features and the most important interactions (rather than all feature importance or significant interactions); the paper seems to merely focus on computing all the significant interactions rather than the most important ones, and in explainability, this is not essentially sought and seems impractical due to many nonzero interactions."
>
> A: We could not make sense of this comment. To be clear, this paper aims to accomplish exactly what you state is the essential goal of explainability. Please feel free to clarify.
>
>
> **Note on Clarity and Narrative**: You have mentioned that you felt that this work was "difficult to follow and does not have a clear storyline". This comment was rather surprising to us, as we have received contradictory feedback from others (including both of the other reviews). We believe the paper is organized in such a way as to naturally explain the different parts of the algorithm, and we have included many figures to aid in explanation. If you have any concrete suggestions on how to improve clarity and the narrative, we welcome it.
>
> **Extending beyond binary functions**: The Mobius transform can certainly be extended beyond the binary inputs, but there remains significant work to extend this algorithm to the $q$-ary case. Critically, we have used group testing designs and results from group testing. Results in non-binary group testing are very limited. In addition, it is unclear how one would extend other core parts of the algorithm, such as the sampling, to the $q$-ary case for $q> 2$.

---

> > ### Comment · Reviewer_vNmZ · 2024-08-11
> > **raise score**
> >
> > I appreciate the responses of the authors to all reviewers and adjust my score accordingly.
> >
> > Just to elaborate more on point 4; the proposed method finds all K significant interactions, which is very interesting since the search space for finding such interactions is exponential. However, the limitation of the method is when K is large (which is typical of the way we define explanation games in explainability), and many of them are not probably of use since we mainly seek the most significant interactions for explanations (e.g., k most significant interactions where k <<<< K).

---

> > > ### Author Response · Authors · 2024-08-11
> > >
> > > Dear Reviewer,
> > >
> > > Thank you for your reply. We wanted to ask:
> > > *What are the issues that still keep this paper from being a clear accept*?
> > >
> > > It would be great if we could resolve these issues and have a unanimous opinion on this paper.
> > >
> > > **Regarding Point 4**:
> > >
> > > Thank you for your clarification. As we now understand it , we believe we have a solution to this. In fact, this solution is *already present* in the paper from line 516-521. The core idea is that we include a post-processing step that takes the $K$ coefficients, and further sparsifies them to our desired level. This is what we have done in Fig. 6 to produce the results for SMT. Thus you can extract a much smaller number $k << K$ coefficients using the following idea in practice:
> > >
> > > 1. Choose $b$ (which serves as a proxy for $K$) and run the full SMT algorithm to get $\hat{F}$. $b$ should be set such as the computational budget allows.
> > >
> > > 2. Apply regression and LASSO steps to further process the transform $\hat{F}$ to reduce the number of coefficients.
> > >
> > > Empirically, we observe that as we increase the amount of regularization in the LASSO, some of the coefficients get set to zero, and their mass is moved to other coefficients. *If this is the main issue you still have, we could include this idea and discussion more prominently in the manuscript*. For your convenience, we have included the relevant appendix excerpt below
> > >
> > > ---
> > > We run SMT to obtain a sparse Möbius representation $\hat{F}$ with support $\mathrm{supp}(\hat{F})$. Then, we fine-tune the values of the coefficients by solving the following regression problem over a uniformly sampled set of points $\mathcal{D} \subseteq \mathbb{Z}\_2^n$:
> > >
> > > \begin{align*}
> > >     & \min\_{\hat{f}, \boldsymbol{\alpha}} \;  \sum\_{\mathbf{m} \in \mathcal{D}} (\hat{f}(\mathbf{m}) - f(\mathbf{m}))^2 \\\\
> > > & \\;\\; \text{s.t.} \quad \hat{f}(\mathbf{m}) = \sum\_{\mathbf{k} \leq \mathbf{m}, \mathbf{k} \in \mathrm{supp}(\hat{F})} \alpha_{\mathbf{k}}, \forall \mathbf{m}.
> > > \end{align*}
> > >
> > > To measure the faithfulness captured by sparse approximations, we modify the regression problem by adding an $\ell_1$ penalty on the values of the Möbius coefficients. Then, we vary the penalty coefficient $\lambda$ to obtain different levels of sparsity:
> > > \begin{align*}
> > >     & \min\_{\hat{f}, \boldsymbol{\alpha}} \;  \sum\_{\mathbf{m} \in \mathcal{D}} (\hat{f}(\mathbf{m}) - f(\mathbf{m}))^2 + \lambda \sum\_{\mathbf{k} \in \mathrm{supp}(\hat{F})} |\alpha_{\mathbf{k}}|\\\\
> > > & \\;\\; \text{s.t.} \quad \hat{f}(\mathbf{m}) = \sum_{\mathbf{k} \leq \mathbf{m}, \mathbf{k} \in \mathrm{supp}(\hat{F})} \alpha_{\mathbf{k}}, \forall \mathbf{m}.
> > > \end{align*}

---

### Official Review · Reviewer_kQft · 2024-07-10

**Soundness:** 3
**Presentation:** 3
**Contribution:** 3
**Rating:** 7
**Confidence:** 3

**Summary:**

This study proposes a new efficient method to compute the Möbius Transform. To understand the behavior of large complex machine learning models, various studies have used game theory concepts such as the Shapley value to measure the impact of input variables on the model's outputs. While Shapley values only focus on individual impacts of input variables, Möbius Transform can score any subset of input variables, which can take into account higher-order interactions between input variables. One of the fundamental challenges of computing Möbius Transform is its exponential computational cost (i.e., requiring $2^n$ inferences with $n$ input variables). This study reduces its cost; with a sparsity assumption, the computation runs with $O(n)$ number of samples, and with a low-degree assumption, it further reduces to $O(\log(n))$. The proposed method (Sparse Möbius Transformer; SMT) exploits a connection between the Möbius Transform in the original $n$-dimensional space (say, F-space) and that in a lower $b$-dimentional space (say, U-space). If the *aliasing set* is a singleton, then the Möbius Transform in F-space can be recovered from that in U-space. The authors proposed methods to detect singletons and turn mulitons into singletons using graph peeling. The experimental results justify the efficiency and the high faithfulness recovered by SMT.

**Strengths:**

- This study offers an efficient computational method of Möbius Transform, which encompasses widely used game-theoretic metrics such as Shapley values and Benzhaf values, and the accelaration from baselines is significant.
- The proposed method is theoretically supported.
- While the paper is densely theoretical, the paper writing is generally easy to follow.
- While not strongly practical and large scale, the assumptions and claims are justified by numerical experiments at key points.

**Weaknesses:**

**Major comments**
- The proposed algorithm works on a $b$-dimensional space, but I was not able to find out how $b$ is set in the experiments and ablation experiments on this hyper-parameter. While the theory suggests $b = O(\log(K))$, this is not very suggestful due to the lack of the constant factor.
- The baselines in Fig. 6 are all first-order metrics. In machine learning and computer vision, second-order metrics (e.g., Interactions) are also popular. Why were those methods not included in the experiments?

**Minor comments**
- It'd be better to clarify several notations, such as $|\mathbf{k}|$ and $[K]$.
- To better convince readers of the connection between Shapley value and Möbius basis, it'd be better to provide a deriviation of the left equality of Eq. (2).
- In the main text, $C$, the number of samplings (?), appears a bit out of sudden at line 193.

**Questions:**

I'd like the Authors to answer the weaknesses raised above. Plus,
- At Limitations, the Authors mentioned that "we may be more interested in taking a sparse projection onto a subset of low-order terms", but I'm not fully following this. Can't we resolve it simply by zeroing out the coefficients of high-order terms (just as "a low-pass filter")?

**Limitations:**

Limitations are adequately presented. For the potential improvements, see Weakness and Questions.

---

> ### Author Rebuttal · Authors · 2024-08-05
>
> 1. $b = O(\log(K))$. In the appendix, we consider the constant $\eta = \frac{2^b}{K}$, which we refer to as the inverse load factor. Theoretically, if we carry through the density evolution analysis, it is possible to find the minimal $\eta$ to ensure convergence of the message passing asymptotically. This is a relatively small number. for instance, $\eta = 0.4073$ suffices for $C=3$ (see [1], Table 1 for a full analysis). In practice, for the real, finite-regime experiments we find $b = \mathrm{ceil}(\log(K))$ typically works well, though it is possible to tune the parameter and adjust $b$, which can be helpful, especially in noisy settings, and if we don't know $K$ a priori, we can increase $b$ until the the algorithm works. (In fact, it is possible to re-use data from smaller values of $b$ to re-run with larger $b$ to expedite this process)
>
> [1] Li, Xiao, Joseph K. Bradley, Sameer Pawar, and Kannan Ramchandran. "SPRIGHT: A fast and robust framework for sparse Walsh-Hadamard transform." arXiv preprint arXiv:1508.06336 (2015).
>
> 2. Fig. 5 focuses on comparisons with second order methods, and reveals the massive computational advantage of our method as compared to those baselines.
> In Fig. 6, we have chosen to focus on first order comparisons because these experiments are run at a reasonably large scale where running higher order methods begins to get computationally difficult. It is likely that higher order methods would perform closely to our methods, just with an increased computational burden.
>
> Question about projection: In this case truncation is not the same thing as a sparse projection onto low order terms, though it is not obvious. The reason this is not the case is because the Mobius transform is a representation in terms of an AND function basis. In fact, the Shapley value, and other interaction indices are usually different projections into the Mobius AND basis.
> This is different from a Fourier transform, which uses an XOR basis. In the case of the Fourier transform, due to orthonormality, your assertion is correct, and simply "zeroing-out" higher order terms is a projection.

---

> ### Comment · Reviewer_kQft · 2024-08-08
> **Response to rebuttal**
>
> Thank you for the clarification. The authors addressed my concerns properly. About 2), while Figure 5 compares second-order methods, it does not show how faithfulness increases with the increase of computational budgets. Practically, we may be interested in the cost of reasonable reconstruction (e.g., 90%). I believe this kind of analysis and second-order version of Fig.6 (with a small dataset) will improve the completeness of this work (just a comment, not asking for an immediate response).
>
> Overall, I'll maintain my score for now. There seems to be a correction/improvement in the proofs pointed out by Reviewer jcxh, so my final score can be affected by the consequences of the discussion between the Authors and Reviewer jcxh.

---

> > ### Author Response · Authors · 2024-08-11
> >
> > Thank you for your prompt response. Our discussions with reviewer jcxh are now concluded. If you have any further comments please let us know.
> >
> > Regarding your comments about comparison, we will keep this in mind as we prepare the final manuscript if this paper is accepted.

---

### Official Review · Reviewer_jcxh · 2024-07-12

**Soundness:** 3
**Presentation:** 3
**Contribution:** 3
**Rating:** 7
**Confidence:** 5

**Summary:**

This paper proposes an algorithm to efficiently compute Mobius transform under the assumption that the function to be transformed is composed of sparse and low-degree interaction terms. The paper also provides an asymptotic analysis of the sample complexity, time complexity, and accuracy of the algorithm.

**Strengths:**

1.	This paper focuses on an important problem in Explainable AI, i.e., how to efficiently compute the Mobius transform (or equivalently, the Harsanyi dividend [cite1]) of a function to obtain the interaction between different input variables. The computation requires inferences on $2^n$ different subsets of input variables, which is intractable in general. The paper proposes an algorithm that leverages the sparse and low-degree structure of the interactions observed in experiments, thus reducing the computational cost.

2.	The algorithm is presented in full detail, with straightforward examples to illustrate each step.

**Weaknesses:**

I have been studying explainable AI (XAI) for years, with a special interest in the Shapley value, the Shapley interaction index, and the Mobius transform. I am impressed by this work and greatly appreciate the progress in the efficient computation of Mobius transform. The sparsity and low-degree structure of interactions and the proposed algorithm can be of great interest to the XAI community, because they help explain a neural network’s decision-making logic into sparse symbolic interactions.

However, I find that the claim and corresponding proof on the singleton detection algorithm are incorrect, and the literature review is insufficient. Nevertheless, **I would like to significantly raise my score to acceptance if these concerns are appropriately addressed.**

1. The most important part of the proposed method is the singleton detection algorithm (in Section 4 and Appendix C.7). However, I find that the claim “singleton identification and detection can be performed without error in the noiseless setting” and the corresponding proof (in Appendix C.7.1) are incorrect. To be specific, the proposed method uses $U(j)$ to efficiently compute of the Mobius transform if $U(j)$ passes the singleton detection, but I find that the singleton detection algorithm in the paper cannot guarantee that $U(j)$ *is indeed a singleton*, although the paper claims that it can.

 To this end, we can easily find counterexamples on which the singleton detection algorithm fails. In summary, we can prove that the algorithm will fail under the following conditions. Given the number of input variables $n$ and the subsampling parameter $b<n$, the interactions satisfy (1) $\forall k\in \\{0,1\\}^n$ s.t. the first $b$ elements of $k$ are all 1’s, $F(k)=-\delta$ if $|k|=n$, $F(k)=\frac{\delta}{n-1-b}$ if $|k|=n-1$, and $F(k)=0$ if $|k|<n-1$, where $\delta \in \mathbb{R}$ is an *arbitrary* scalar (2) $\forall k\in \\{0,1\\}^n$ s.t.  at least one of the first $b$ elements of $k$ is 0, the value of $F(k)$ can be *arbitrary*.

**Counterexample:** Let us consider $n=6$ input variables and set $b=2$ for subsampling, which follows the setting in Figure 3. Suppose there are five non-zero interactions in total: $k_1=110111, k_2=111011, k_3=111101, k_4=111110, k_5=111111$. And $ F(k_1)= F(k_2)= F(k_3)= F(k_4)=1/3$, $F(k_5)=-1$. In the $c$-th subsampling run, let the subsampling matrix $H_c=[[1,0,0,0,0,0],[0,1,0,0,0,0]]$. Then, we will obtain $u_c(00)=f(001111)$, $u_c(01)=f(011111)$, $u_c(10)=f(101111)$, $u_c(11)=f(111111)$, and also $U_c(00)=0$, $U_c(01)=0$, $U_c(10)=0$, $U_c(11)=F(k_1)+ F(k_2) + F(k_3) + F(k_4) + F(k_5)$.
The next step in the proposed method is to detect if each $U_c(j)$ is a zeroton, a singleton, or a multiton. This is done by adding different unit vectors $d_{c,p}=e_p, p=1,…,n$, and obtain $u_{c,p}(l)=f\left(\overline{H_c^\top \bar{l} + d_{c,p}}\right) \rightarrow U_{c,p}(j)=\sum_{k\le \bar{d} \ s.t. Hk=j} F(k)$. We list the results of $U_{c,p}(j)$ for $j=11$, in the following table:

|              | p=0    | p=1    | p=2    | p=3    | p=4    | p=5    | p=6    |
| ------------ | ------ | ------ | ------ | ------ | ------ | ------ | ------ |
| $d_{c,p}$            | 000000 | 100000 | 010000 | 001000 | 000100 | 000010 | 000001 |
| $U_{c,p}(11)$ | 1/3    | 0      | 0      | 1/3    | 1/3    | 1/3    | 1/3    |

**The above counterexample passes the singleton detection algorithm, but $U_c(11)= F(k_1)+ F(k_2) + F(k_3) + F(k_4) + F(k_5)$ contains five terms and is apparently not a singleton.** To be more specific, we can see that the metric $y_{c,p}=1-\frac{U_{c,p}(11)}{U_{c,0}(11)}$ is either 0 or 1, for $p=1,…,n$. Therefore, according to the criteria in Equation (9), the multiton $U_c(11)= F(k_1)+ F(k_2) + F(k_3) + F(k_4) + F(k_5)$ will be mistakenly judged as a singleton by the algorithm. As a result, the algorithm will yield an incorrect output that contains only one interaction term, while the ground truth contains five interaction terms.

I suggest the authors either revise the claim and the proof in Appendix C.7.1 to be in an asymptotic or probabilistic manner, or provide detailed discussion to the failure cases of the singleton detection algorithm. Even though the algorithm cannot handle all scenarios, I’m still willing to raise my score if proper discussion on the failure cases is added.

---

2. The literature review is insufficient. The core background behind the proposed method is the sparsity of Mobius transform. In fact, the sparsity of Mobius transform (or equivalently, the Harsanyi dividend) has been well studied by many previous works, but these works are not properly discussed in the paper. For example, [cite2, cite3] observed the sparsity of Mobius transform on a wide range of network architectures (e.g., MLPs, CNNs, transformers, PointNet) trained on various tasks (e.g., tabular data classification, image recognition, sentiment analysis, pointcloud classification). [cite4] explored the source of the sparsity and even proved the sparsity of Mobius transform under three sufficient conditions. The authors are expected to include these previous works in the Related Work section or in the discussion of the sparsity assumption in Section 2.

---

3. The paper reports the relationship between Mobius transform and the Shapley value in Equation (2) and Appendix A. However, this relationship has already been discussed in the original paper of Harsanyi dividend [cite1]. Therefore, I encourage the authors to cite paper [cite1] as a proper reference.


[cite1] John C. Harsanyi. A simplified bargaining model for the n-person cooperative game. International Economic Review, 4(2):194–220, 1963.

[cite2] Ren et al. Defining and Quantifying the Emergence of Sparse Concepts in DNNs. CVPR, 2023.

[cite3] Li and Zhang. Does a Neural Network Really Encode Symbolic Concepts? ICML, 2023.

[cite4] Ren et al. Where We Have Arrived in Proving the Emergence of Sparse Interaction Primitives in DNNs. ICLR, 2024.

**Questions:**

See weaknesses above

---

After rebuttal: The authors' response mostly addresses my concern about the singleton detection algorithm. The authors also commit to incorporating the relevant literature into their paper. Therefore, I have raised my score to acceptance.

---

> ### Author Rebuttal · Authors · 2024-08-05
>
> Dear reviewer jcxh. Thank you for your through and helpful review. We are delighted that you are impressed with our contributions towards efficient Mobius Transform computation, and agree with our assumptions of low-degree interactions and sparsity. We believe that our work will in time be appreciated as a significant step forward for Mobius transform computation.
>
> **Proof Correction**
>
> We have corrected the technical issue you highlighted, and will include that information in a separate comment.
>
> **Literature Review**
>
>  Regarding the literature review, we are very happy to see these other references, particularly the modern ones. It is exciting to see a community of people working on these types of problems, and these are excellent examples that highlight the practical points of our assumptions. These papers will bolster our current Section 2, and add credibility to our work. We will also certainly include the Harsanyi reference.
>
> **Final note**
>
>  Reviewer vNmZ has raised concerns about the validity of our assumptions for practical settings. We have made the case that these are in fact interesting and important assumptions to consider. Given your expertise in this area, we would welcome your contribution to the discussion.

---

> ### Author Response · Authors · 2024-08-05
> **Proof Revision (Assumption)**
>
> **Updated Proof**
>
> We would like to thank the reviewer for identifying this issue. After a thorough analysis, we see that the previous proof in Appendix C.7.1 (which applies to the noiseless case only) was only for multitons of order $2$, and pathological examples as you described can be constructed for higher order multitons. We have resolved this issue with the full proof included below. As the reviewer has suggested, we have introduced a very mild probabilistic assumption on the values of the non-zero coefficients, so that they may not be completely arbitrary.
>
>
> The proof for our noisy result, which uses a more complicated approach and requires additional samples (along with already restricting $F(\mathbf{k})$) is not impacted by this issue.
>
> **Assumption 2.3 (No Cancellation)** Suppose the non-zero values $F(\mathbf{k}_1)$, ... , $F\(\mathbf{k}_K\)$  are sampled from a joint distribution $\mathbb{P}$ that satisfies the following condition:
>
> \begin{equation}
>     \sum_{i \in S}F(\mathbf{k}_i) \neq 0, \;\forall S \subseteq [K], \; S \neq \emptyset.
> \end{equation}
>
>
> Assumption 2.3 is quite mild, and there are many classes of $\mathbb{P}$ that satisfy this assumption. *For example, any absolutely continuous $\mathbb{P}$ satisfies Assumption 2.3 a. s*.
>
> **Proof.**
>
> For simplicity let $\mathbf{F}$ represent a $K$ dimensional random vector containing $F\( \mathbf{k}\_i \)$ at  index $i$.
> Let the set $\mathcal{R}(S, \alpha) = \\{\mathbf{F} : \sum_{i \in S}F(\mathbf{k}\_i) = \alpha\\}$. Since $\mathbb{P}$ is absolutely continuous, a density $p$ exists such that:
>
> \begin{equation}
>     \mathbb{P}(\mathbf{F} \in \mathcal{R}(S,\alpha))  = \int_{\mathcal{R(S,\alpha)}}dp.
> \end{equation}
>
> However, $\mathrm{dim}(\mathcal{R}(S,\alpha)) = K - \left\lvert S \right\rvert$. Thus for $S \neq \emptyset$, $\mathcal{R}(S,\alpha)$ has Lebesgue measure zero, thus
>
> \begin{equation}
>    \mathrm{Pr}\left(\sum_{i \in S} F(\mathbf{k}_i) = 0\right) =  \mathbb{P}(\mathbf{F} \in \mathcal{R}(S,0)) =  \int\_{\mathcal{R}(S,0)} dp = 0.
> \end{equation}

---

> ### Author Response · Authors · 2024-08-05
> **Proof Revision (Proof, Theorem 1)**
>
> We separate the proof into three parts:
>
> (1) Prove $\mathrm{Detect}(\mathbf{U}_c(\mathbf{j})) = \mathcal{H}_Z \implies \mathrm{Type}(\mathbf{U}_c(\mathbf{j})) = \mathcal{H}_Z$.
>
> (2) Prove $ \mathrm{Detect}(\mathbf{U}_c(\mathbf{j})) = \mathcal{H}_S \implies \mathrm{Type}(\mathbf{U}_c(\mathbf{j})) = \mathcal{H}_S$.
>
> (3) Prove $\mathrm{Detect}(\mathbf{U}_c(\mathbf{j})) = \mathcal{H}_M \implies \mathrm{Type}(\mathbf{U}_c(\mathbf{j})) = \mathcal{H}_M$.
>
> Consider the subsampling group $\mathbf{U}_{c}(\mathbf{j})$ for some fixed $c, \mathbf{j}$. We first consider the case where $\left\lvert \mathbf{k}\right\rvert$ is not restricted, and denote the set of non-zero indices $\mathbf{k}_1, \dotsc, \mathbf{k}_K$ as $\mathcal{N}$.
>
> **Proof of (1)**
>
> Let $\mathrm{Detect}(\mathbf{U}\_c(\mathbf{j})) = \mathcal{H}\_Z$, and for contradiction's sake assume $\mathrm{Type}(\mathbf{U}\_c(\mathbf{j})) \neq \mathcal{H}\_Z$.
> \begin{equation}
>         \mathrm{Detect}(\mathbf{U}\_c(\mathbf{j}))  = \mathcal{H}\_Z \implies \sum_{\mathbf{k} \leq \bar{\mathbf{d}}\_p\; \text{ s.t. } \mathbf{H}\_c \mathbf{k} = \mathbf{j}} F(\mathbf{k}) = 0 \;\forall p,
> \end{equation}
> Since $\mathrm{Type}(\mathbf{U}_c(\mathbf{j})) \neq \mathcal{H}_Z$, we have that $\mathcal{N} \cap \\{ \mathbf{k} : \mathbf{H}_c \mathbf{k} = \mathbf{j}\\} \neq \emptyset$.
> Thus, if we consider  the above implication for the case of $p=0$ and noting that $\mathbf{d}_0 = \boldsymbol{0}$, we have:
> \begin{equation}
>     \mathrm{Detect}(\mathbf{U}\_c(\mathbf{j}))  = \mathcal{H}\_Z \implies \sum\_{\mathcal{N} \cap \\{ \mathbf{k} : \mathbf{H}\_c \mathbf{k} = \mathbf{j}\\}} F(\mathbf{k}) = 0
> \end{equation}
> But considering the no cancellation condition, this is impossible, thus proving (1).
>
> **Proof of (2)**
>
> Note that the converse of (1) is true (the proof is immediate). Thus, proving $ \mathrm{Detect}(\mathbf{U}\_c(\mathbf{j})) = \mathcal{H}\_S \implies \neg(\mathrm{Type}(\mathbf{U}\_c(\mathbf{j})) = \mathcal{H}\_M$) is the same as proving (2).
> We will again use the method of contradiction, and assume $ \mathrm{Detect}(\mathbf{U}\_c(\mathbf{j})) = \mathcal{H}\_S$ and $\mathrm{Type}(\mathbf{U}\_c(\mathbf{j})) = \mathcal{H}\_M$.
> \begin{equation}
>         \mathrm{Detect}(\mathbf{U}\_c(\mathbf{j}))  = \mathcal{H}\_S \implies U_{c,p}(\mathbf{j}) \in \\{0, U\_{c,0}(\mathbf{j})\\}  \; \forall p > 1.
> \end{equation}
> Note that by our assumption, $U_{c,0}(\mathbf{j}) \neq 0$. By our assumption that $\mathrm{Type}(\mathbf{U}_c(\mathbf{j})) = \mathcal{H}_M$, we must have $\left\lvert \mathcal{N} \cap \{ \mathbf{k} : \mathbf{H}_c \mathbf{k} = \mathbf{j}\} \neq \emptyset \right\rvert \geq 2$.
> Choose $\mathbf{k}\_1, \mathbf{k}\_2 \in \mathcal{N} \cap \\{ \mathbf{k} : \mathbf{H}\_c \mathbf{k} = \mathbf{j}\\}$.
> Given our choice of $\mathbf{d}\_p$ $\exists p^* > 1$ s.t. only *one* of $\mathbf{k}\_1$ or $\mathbf{k}\_2 \leq \bar{\mathbf{d}}\_{p^*}$.
> Without loss of generality we will assume $\mathbf{k}\_2 \leq \bar{\mathbf{d}}\_{p^*}$ and
> $\mathbf{k}\_1 \nleq \bar{\mathbf{d}}\_{p^*}$. Now, define the following sets:
> \begin{eqnarray}
>     \mathcal{J}\_0 &=& \mathcal{N} \cap \\{ \mathbf{k} : \mathbf{H}\_c \mathbf{k} = \mathbf{j}\\} \\\\
>     \mathcal{J}\_{p^*} &=& \mathcal{N} \cap \\{ \mathbf{k} : \mathbf{H}\_c \mathbf{k} = \mathbf{j} \; \mathbf{k} \leq \bar{\mathbf{d}}\_{p^*} \\}
> \end{eqnarray}
> We know $\mathbf{k}\_2 \in \mathcal{J}\_{p^*}$ and $\mathbf{k\_1} \in \mathcal{J}\_0\setminus \mathcal{J}\_{p^*}$, thus $\left\lvert \mathcal{J}\_{p^*}\right\rvert \geq 1$ and $\left\lvert \mathcal{J}\_0\setminus \mathcal{J}\_{p^*}\right\rvert \geq 1$. With this, we can show that the implication above cannot be satisfied.
>
> *Case 1:* $U_{c,p^*}(\mathbf{j}) = 0$.
>
> \begin{equation}
>     U_{c,p^*}(\mathbf{j}) = 0 \implies \sum_{\mathbf{k} \in \mathcal{J}_p^*} F(\mathbf{k}) = 0.
> \end{equation}
> Since $\mathcal{J}_p^*$ is not empty, from our distributional assumption, the above sum cannot be $0$.
>
> *Case 2:* $U_{c,p^*} = U_{c,0}(\mathbf{j})$.
> \begin{equation}
>         U_{c,0}(\mathbf{j}) - U_{c,p^*}(\mathbf{j}) = 0 \implies \sum_{\mathbf{k} \in \mathcal{J}_0 \setminus \mathcal{J}_p^*} F(\mathbf{k}) = 0.
> \end{equation}
> Since $\mathcal{J}_0 \setminus \mathcal{J}_p^*$ is not empty, from our distributional assumption, the above sum cannot be $0$. This implies that $\mathrm{Type}(\mathbf{U}_c(\mathbf{j})) = \mathcal{H}_M$ must be false, thus proving (2).
>
> **Proof of (3):**
>
> Since we have a converse for (1), a converse for (2) suffices to prove (3). The converse follows below:
> \begin{eqnarray}
>  \mathrm{Type}(\mathbf{U}\_c(\mathbf{j})) = \mathcal{H}\_S \implies \exists \mathbf{k}^* \text{ s.t. } U\_{c,p} \in \\{0, F(\mathbf{k^*})\\},\;\forall p.
> \end{eqnarray}
> Since $F(\mathbf{k^*}) \neq 0$, we have $U_{c,0}(\mathbf{j}) \neq 0$ and all entries of $\mathbf{U}_{c}(\mathbf{j})$ are either $F(\mathbf{k^*})$ or $0$. Thus,  $\mathrm{Detect}(\mathbf{U}_c(\mathbf{j})) = \mathcal{H}_S$.

---

> ### Author Response · Authors · 2024-08-05
> **Proof Revision (Proof, Theorem 2)**
>
> Here we include a sketch of the argument for the noiseless part of Theorem 2, which focuses on the case $\left\lvert \mathbf{k}\right\rvert \leq t$ (the rest of Theorem 2 is unchanged).
>
> (1) Prove $\mathrm{Detect}(\mathbf{U}\_c(\mathbf{j})) = \mathcal{H}\_Z \implies \mathrm{Type}(\mathbf{U}\_c(\mathbf{j})) = \mathcal{H}\_Z$.
>
> (2+3) Prove $\mathrm{Pr}\left(\mathrm{Detect}(\mathbf{U}\_c(\mathbf{j})) = \mathrm{Type}(\mathbf{U}\_c(\mathbf{j})) \right) \rightarrow 1$.
>
> **Proof of (1)**
>
> The proof is identical to above.
>
> **Proof of (2+3)**
>
> The proof is the same, with one notable exception. In the low-degree case, $p^*$ may not always exist. Using the results in citation [47] from the paper, and the union bound that already exists in the proof in Appendix C.7.2, we show that $p^*$ exists with probability approaching $1$.

---

> ### Comment · Reviewer_jcxh · 2024-08-08
>
> Thank you for your response. The added assumption and proof address my concern for the singleton detection algorithm. I hope this assumption and proof, as well as the mentioned literature review for the sparsity of Mobius transform, will be properly stated in the final version of the paper if the paper is accepted. Furthermore, I have raised my score accordingly.

---

> > ### Author Response · Authors · 2024-08-11
> >
> > Thank you for your diligence as a reviewer, we will be certain to correctly state the theorems with this additional assumption, and include this revised proof in the appendix.

---

### Author Rebuttal · Authors · 2024-08-06

Dear Reviewers, ACs and Senior ACs.

We have posted individual rebuttals to all three reviewers. We believe we have addressed the concerns of all reviewers. We would particularly like to thank jcxh as an outstanding reviewer.

Taking into account the issues, which are now resolved, we are still confident in the candidacy of this manuscript, and look forward to the discussion period.

---

### Decision · Program_Chairs · 2024-09-25

**Decision:**

Accept (poster)

**Comment:**

As the reviews and the follow-up discussion illustrate, this is an interesting contribution towards the theory and efficient algorithms for computing the Mobius transform that can be used to interpret which tokens in the input were the ones most responsible for the final output. The paper provides interesting connections with group testing and is technically solid (modulo a correction in one of the proofs pointed out in the review process). I recommend acceptance as a poster.